# Non-calyceal inputs gate the timing of calyx of Held evoked MNTB output
Laura Console-Meyer[1,2], Florian Jenzen[1], Nikolaos Kladisios ⓘ[1] & Felix Felmy ⓘ[1] ✉

Synaptic filtering by short-term plasticity (STP) influences the strength and timing of the input-output function of neurons during ongoing activity. In the medial nucleus of the trapezoid body (MNTB), an ultrafast sign-inverting relay, short-term depression (STD) of the dominant calyx of Held inputs affects the exquisite output timing of these neurons and thereby influences sound processing relevant for sound source localization. We find that additional excitatory non-calyceal inputs, targeting soma and dendrites of MNTB neurons, show prominent frequency-dependent facilitation and asynchronous release leading to a steady depolarizing conductance. The integration of non-calyceal and the calyx of Held input provides a mechanism for adjusting the temporal precision and faithfulness of output during prolonged high-frequency activity in a range relevant for auditory processing. Thus, non-calyceal inputs tune suprathreshold output generation in a functionally relevant manner, qualified to set the neuronal representation of sound sources in space.

Synaptic short-term plasticity (STP) is evident as depression, facilitation or their interplay[1]. Short-term depression (STD) can be driven by pool depletion, calcium current inactivation, or postsynaptic receptor desensitization[2]. Short-term facilitation (STF) is at least based on residual calcium[1,3], calcium buffer saturation[4–6], calcium current facilitation[7], and supported by receptor diffusion[8]. STP impacts ongoing input-output functions in neurons by generating filter functions, gain control, and affecting temporal integration[9–12]. For a single neuron, different inputs might have opposing STPs. Therefore, their combined integration can compensate for individual STP forms of a specific input or produce novel filter-functions for the neuron's output generation.

The STP function in temporal integration is highlighted at neurons in the superior olivary complex (SOC)[9,11]. The binaural detectors in the medial and lateral superior olive (MSO and LSO) are biophysically tuned to detect coincidental arrival in the sub-millisecond time range[13]. This ultra-fast integration is matched to the physiological detection of sound sources in the azimuthal plane of mammals[13] and inhibition adjusts this temporal precision[14,15]. This relevant inhibition is fed forward from the medial nucleus of the trapezoid body (MNTB) with exquisite temporal precision[16,17]. This inhibition also projects beyond the SOC to various additional nuclei along the ascending auditory pathway[18] to adjust other spectral and binaural filters. Besides the MNTB's rapid feed-forward inhibition to the binaural detectors in the SOC, local feedback inhibition controls the neuronal representation of the sound sources and thus space[19,20].

The temporal precision of MNTB neurons is highlighted by phase locking their suprathreshold output to sound stimulations[21–24]. These phase-locking abilities are based on a set of voltage-gated potassium channels[25–27] and the somatic, glutamatergic calyx of Held synapse. This synapse triggers large, fast excitatory postsynaptic currents (EPSCs) that show use-dependent STD[28,29]. Despite the STD, this synapse leads to high-fidelity supra-threshold output, but with increasing failures and loss of timing during prolonged activation[21,22,28,30,31]. Next to the dominating calyx of Held, MNTB neurons receive synaptic inputs on their dendrites and soma[32,33]. Some of these non-calyceal inputs generate glutamatergic, temporally unprecise responses[34]. Their functional properties including the STP and integration with the calyx of Held input remain elusive. However, the capability of MNTB neurons to integrate different inputs can be inferred from the changed tuning functions after blocking inhibition[35].

Here we characterize the non-calyceal inputs including their STP and show their crucial role in shaping the temporal precision of supra-threshold output generation driven by the calyx of Held in MNTB neurons. We find that non-calyceal inputs are present in mammals of large evolutionary distance, show strong facilitation, and can reduce action potential failures during STD periods of calyx of Held inputs. Moreover, the non-calyceal inputs gate the timing of MNTB output generation in a physiologically relevant time range. Thus, non-calyceal inputs provide support for MNTB neurons action potential success rate and timing and thereby might affect sound processing, including the spatial representation of sound sources.

[1]Institute of Zoology, University of Veterinary Medicine Foundation, Hannover, Buenteweg 17, Hannover, Germany. [2]Hannover Graduate School for Neurosciences, Infection Medicine and Veterinary Sciences (HGNI), Buenteweg 2, Hannover, Germany. ✉e-mail: felix.felmy@tiho-hannover.de

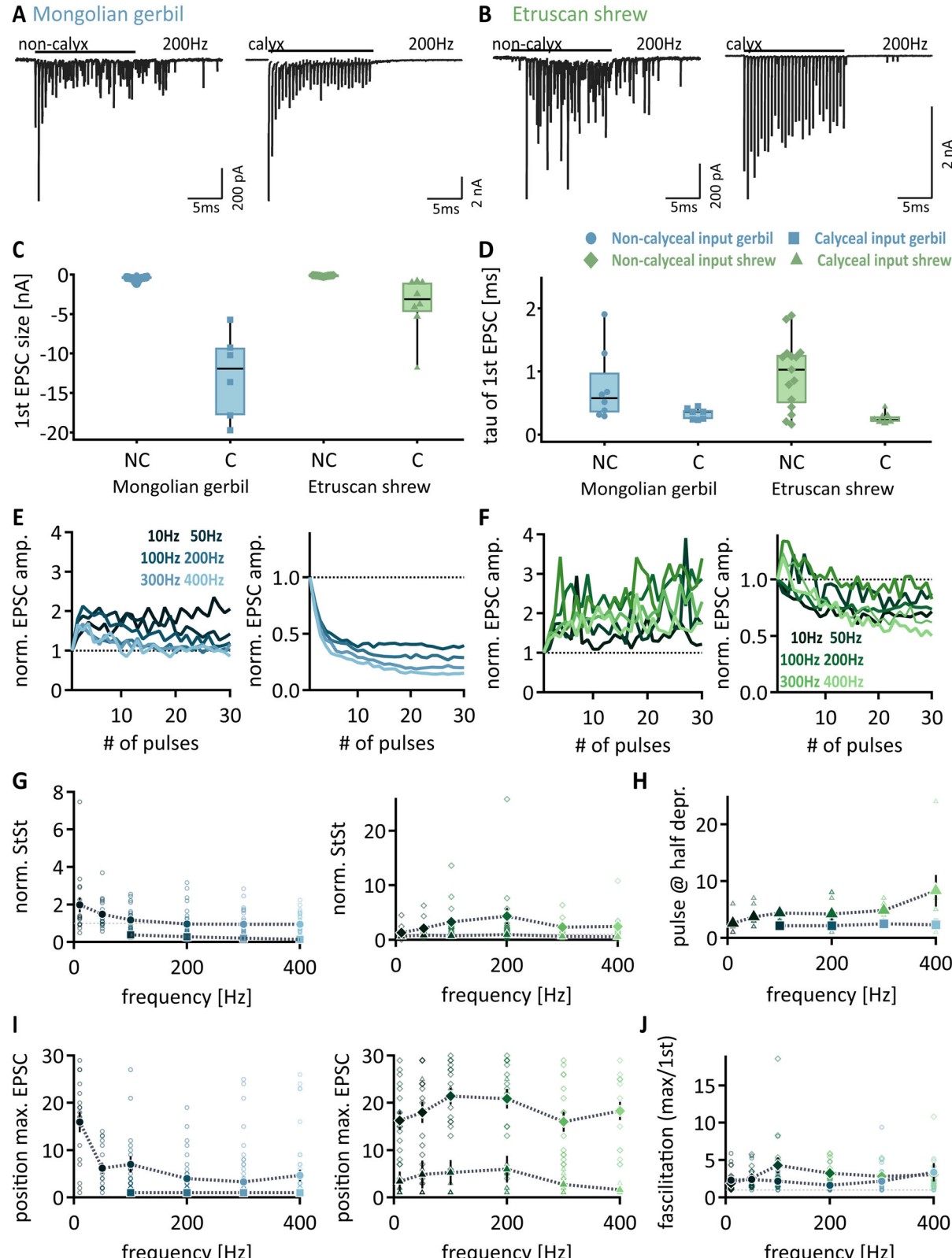

## Results

### Synaptic physiology of non-calyceal excitatory inputs to MNTB neurons

MNTB neurons are innervated by the calyx of Held and excitatory non-calyceal synapses. The STP of non-calyceal inputs was compared to the calyx of Held at near physiological conditions regarding temperature and external

calcium concentration[36] by applying pulse trains of 30 stimulations at different frequencies. Comparative recordings in gerbils and Etruscan shrews were carried out to highlight the general attributes of STP in both synapse types in this archaic neuronal circuit. In both species, non-calyceal inputs rather exhibit STF in contrast to dominating STD of the calyx of Held (Fig. 1A, B), indicating an overall evolutionary stability of STP.

**Fig. 1 | Short-term plasticity of synaptic input on MNTB principal neurons.**
**A, B** Exemplary EPSC responses to 200 Hz stimulation pulse, recorded from
Mongolian Gerbil and Etruscan shrew, respectively. Black line indicates duration of
stimulation pulse. Left: non-calyceal input, right: calyceal input. **C, D** First EPSC size
and decay time compared between input types and species. Blue indicates Mongolian gerbil; green indicates Etruscan shrew. In Gerbil, circled symbols denote non-
calyceal inputs ($n = 26$) and squares indicate calyceal inputs ($n = 6$). In shrew,
rhombuses indicate non-calyceal inputs ($n = 16$) and triangles depict calyx inputs
($n = 8$). Data show single cell values and median ± quartile (black). **E, F** Normalized
EPSC amplitudes to the first response, either in gerbil (blue) or in shrew (green), for
each recorded stimulation frequency (10 Hz, 50 Hz, 100 Hz, 200 Hz, 300 Hz,
400 Hz). The left plot shows non-calyceal input, the right shows calyx of Held
input responses. Values below the dotted line indicate depression, values above
facilitation. **G** Normalized steady state in dependence on stimulation frequency
(mean ± SEM) in either calyceal (squares) or non-calyceal inputs (circles). Left:
gerbil, right: shrew. Single cell data depicted in open symbols and matching color
code. **H** Number of pulses needed to reach half depression in dependence on sti-
mulation frequency. Only calyceal input is shown, as STD is apparent. Data shown as
mean ± SEM. Single cell data depicted in open symbols and matching color code.
**I** Position of maximum EPSC in frequency-dependence in gerbil (left) and shrew
(right) in either non-calyceal data (circles/rhombuses) or calyceal data (squares/
triangles), respectively. Single cell data depicted in open symbols and matching color
code. **J** Facilitation calculated as maximum EPSC divided by the first EPSC of non-
calyceal inputs in gerbil (blue) and shrew (green) plotted against stimulation fre-
quency. Single cell data depicted in open symbols and matching color code. For (**E–J**)
$n = 27$ for non-calyceal and $n = 6$ for calyceal inputs of gerbil MNTB neurons, $n = 16$
and $n = 8$, respectively, for shrew.

The size and decay kinetics of the first EPSC of the stimulation trains
were analyzed. The EPSC size differs significantly between input types in
both species (Fig. 1C; Gerbil: -0.32 nA and -11.9 nA medians for non-
calyceal and calyx of Held, $p < 0.0001$; Shrew: -0.11 nA and -3.1 nA medians
for non-calyceal and calyx of Held, $p < 0.0001$, all statistics and presented
data are available under supplementary data). These differences may
represent an underestimate, as calyx of Held stimulation has been based on a
single fiber input, whereas the non-calyceal EPSCs may be composed of
multiple fibers. Under these recording conditions, the EPSC amplitude of
non-calyceal inputs was smaller in Etruscan shrew compared to gerbils
($p < 0.0001$). Similarly, the EPSC size generated by the calyx of Held was
smaller in Etruscan shrew (Fig. 1C, $p = 0.0047$). Thus, like the difference
between gerbils and bats[28], Etruscan shrews show smaller inputs compared
to gerbils. The decay time of non-calyceal EPSCs was, on average, slower and
more heterogeneous compared to calyx of Held EPSCs in both species
(Fig. 1D; Gerbil: $p = 0.0368$; Shrew: $p = 0.0036$). Between the species, no
differences in EPSC decays were apparent for the different input types (non-
calyceal: $p = 0.4762$; calyx of Held: $p = 0.182$).

STP was quantified in a frequency-dependent manner and normalized
to the first evoked EPSC of the train. Within the stimulated response, the
largest EPSC between pairs of pulses was taken as the evoked EPSC, while
the smaller ones were attributed to asynchronous and not used for STP
analysis. In both species, non-calyceal inputs initially facilitated and showed,
if at all, only little depression at the end of the stimulation train (Fig. 1E, F).
As evident from the recordings, EPSC size of non-calyceal inputs largely
varied during the train response. Moreover, during train stimulations, these
inputs exhibited a substantial increase in asynchronous release, followed by
delayed release after the cessation of the stimulation. The calyx of Held
EPSCs showed depression during the stimulation train (Fig. 1E, F). This
depression was more robust in gerbils compared to Etruscan shrews, where
depression followed an initial facilitation (100 Hz: $p = 0.0029$; 200 Hz:
$p = 0.0321$; 300 Hz: $p = 0.0029$; 400 Hz: $p = 0.0291$).

To quantify STP in more detail, the normalized steady-state level was
determined from the average size of the last five EPSCs. Steady-state analysis
of non-calyceal inputs in gerbils revealed frequency-dependent STP, since
facilitation occurred at low frequencies, while slight depression was
apparent at high stimulation frequencies (Fig. 1G; frequency dependence of
STP: ANOVA $p = 0.0008$). In Etruscan shrew, the steady-state facilitation
was more pronounced and increased in a frequency-dependent manner
(Fig. 1G; ANOVA $p < 0.0001$). For EPSCs mediated by the calyx of Held
input, the steady state depression at high stimulation frequencies was larger
in gerbils compared to Etruscan shrews (Fig. 1G; 100 Hz: $p = 0.0813$; 200 Hz:
$p = 0.0863$; 300 Hz: $p = 0.0482$; 400 Hz: $p = 0.0344$). To further differentiate
the calyx of Held-mediated STP between the species, the depression kinetics
were extracted as the pulse number when half of the maximal depression
was reached (Fig. 1H). For all tested frequencies, depression was reached
faster in gerbils compared to Etruscan shrew (100 Hz: $p = 0.0029$; 200 Hz:
$p = 0.0321$; 300 Hz: $p = 0.0029$; 400 Hz: $p = 0.0291$).

To quantify the facilitation, the pulse number within the train that
elicited the largest EPSCs was extracted (Fig. 1I). In non-calyceal inputs in

gerbils, the maximal response amplitude occurred late during low and early
during high frequency stimulations (Fig. 1I; Friedman: $p = 0.0002$; 10 Hz vs.
200 Hz: $p = 0.0026$; 10 Hz vs 300 Hz: $p = 0.0008$; 10 Hz vs. 400 Hz: $p = 0.02$).
In non-calyceal inputs in shrew, no frequency-dependence was observed
(Friedman: $p = 0.1912$) and the maximal EPSC occurred later during the
stimulation train (Fig. 1I, 10 Hz: $p = 0.9181$; 50 Hz: $p = 0.0002$; 100 Hz:
$p < 0.0001$; 200 Hz: $p < 0.0001$; 300 Hz: $p < 0.0001$; 400 Hz: $p < 0.0001$). A
similar difference in the position of the maximal EPSC response in the
stimulation train was found for calyx of Held responses. In gerbils, the
largest response was usually the first EPSC, while in Etruscan shrew it was on
average at the fourth pulse in the train. The difference in the occurrence of
the largest response was significant at 200 Hz and 300 Hz between both
species (Fig. 1I, gerbil vs Etruscan shrew: 100 Hz: $p = 0.0849$; 200 Hz:
$p = 0.015$; 300 Hz: $p = 0.031$; 400 Hz: $p = 0.0849$). However, within the spe-
cies, no substantial frequency-dependence of the position of the maximal
EPSC occurrence in the train was present (Gerbil: Friedman: $p > 0.9999$;
Shrew: $p = 0.0737$). To determine the relative size of the short-term facil-
itation, the largest EPSC response in the train was normalized to the first
EPSC for non-calyceal inputs. These inputs facilitate at all frequencies tested.
In Etruscan shrew, this facilitation shows frequency dependence (Friedman:
$p = 0.0062$, 10 Hz vs. 200 Hz $p = 0.0049$; 10 Hz vs. 300 Hz $p = 0.0197$). Again,
STF is more prominent at higher frequencies in Etruscan shrew compared to
gerbil (Fig. 1J, 10 Hz: $p = 0.3825$; 50 Hz: $p = 0.5274$; 100 Hz: $p = 0.0168$;
200 Hz: $p < 0.0001$; 300 Hz: $p = 0.0037$; 400 Hz: $p = 0.3047$). Taken together,
despite species-dependent differences, the non-calyceal inputs show facil-
itation while the calyx of Held input is characterized by depression.

So far, our analysis has focused on the phasic release that is tightly
coupled to stimulation frequency. As evident from Fig. 1A, B, in the non-
calyceal synapses, neurotransmitters are released both in a phasic and an
asynchronous manner. Therefore, we next quantify the delayed release
component of the asynchronous release after the cessation of the stimula-
tion. The quantification of delayed release was restricted to the non-calyceal
inputs, since it is more evident compared to the calyx of Held, where
asynchronous and delayed release events have been analyzed before[37,38]. In
both species, the non-calyceal inputs produced pronounced delayed release
that persisted for up to 500 ms after the last pulse of the stimulation trains
(Fig. 2A, B). To probe for use-dependency of the delayed release, the time of
EPSC occurrences were extracted after the phasic response to the last sti-
mulation pulse for a period of 500 ms. When delayed EPSCs did not overlap
but appeared temporally separated, their amplitudes and decay times were
individually extracted (Fig. 2A, B).

We observed a strong heterogeneity of EPSC amplitudes and decay
times within delayed release in single cells in both species (Fig. 2A, B). We
found that in most inputs, a fraction of slower EPSCs occurred, which
tended to be smaller, while the occurring fraction of larger EPSCs were
usually among the fastest (Fig. 2A, B). This heterogeneity might suggest that
the extracted EPSCs either originate from different cell compartments and
undergo dendritic filtering, or the activated synapses contain glutamate
receptors of different subunit compositions. To analyze whether the EPSC
amplitudes and decay times within the delayed release events differ between

**Fig. 2 | Delayed release of non-calyceal inputs in Mongolian Gerbil and Etruscan shrew.**
**A** Exemplary trace of delayed release in gerbil. Black EPSCs mark the last two phasic pulses and are further noted with a star on top. Each event occurring in the post-stimulus time is noted with a stroke on top. Since occurring EPSC events show strong heterogeneity, two exemplary EPSCs are shown below, one as an example of a rather fast and larger EPSC and another as an example of a slower and smaller EPSC. **Ai** EPSC amplitude of occurring delayed events plotted against their EPSC decay time of one exemplary cell. Each circle represents one event. **Aii** Left: Histogram of the average of normalized peak events. The top graph shows events followed by a 10 Hz stimulation and the bottom shows events followed by a 200 Hz stimulation. Right: normalized events were cumulatively added and plotted against post-stimulus time for each frequency. The dashed horizontal line marks 70% of the cumulative events, which were extracted for (**D**). **B** to **Bii** same as (**A–Aii**) but for Etruscan shrew. **C** Left: EPSC amplitude in both species, right: EPSC decay in both species. Gray symbols show the mean per cell ± SD. Colored symbols show the overall species average ± SEM. **D** Left: Average post-stimulus time point at 70% occurrence of cumulative events (mean ± SEM) in both gerbil (blue) and shrew (green), plotted against stimulation frequency. Right: Average ( ± SEM) number of events in both gerbil (blue) and shrew (green), plotted against stimulation frequency. Single cell data (*n* = 16 for gerbil and *n* = 9 for shrew) depicted in open symbols and matching color code.

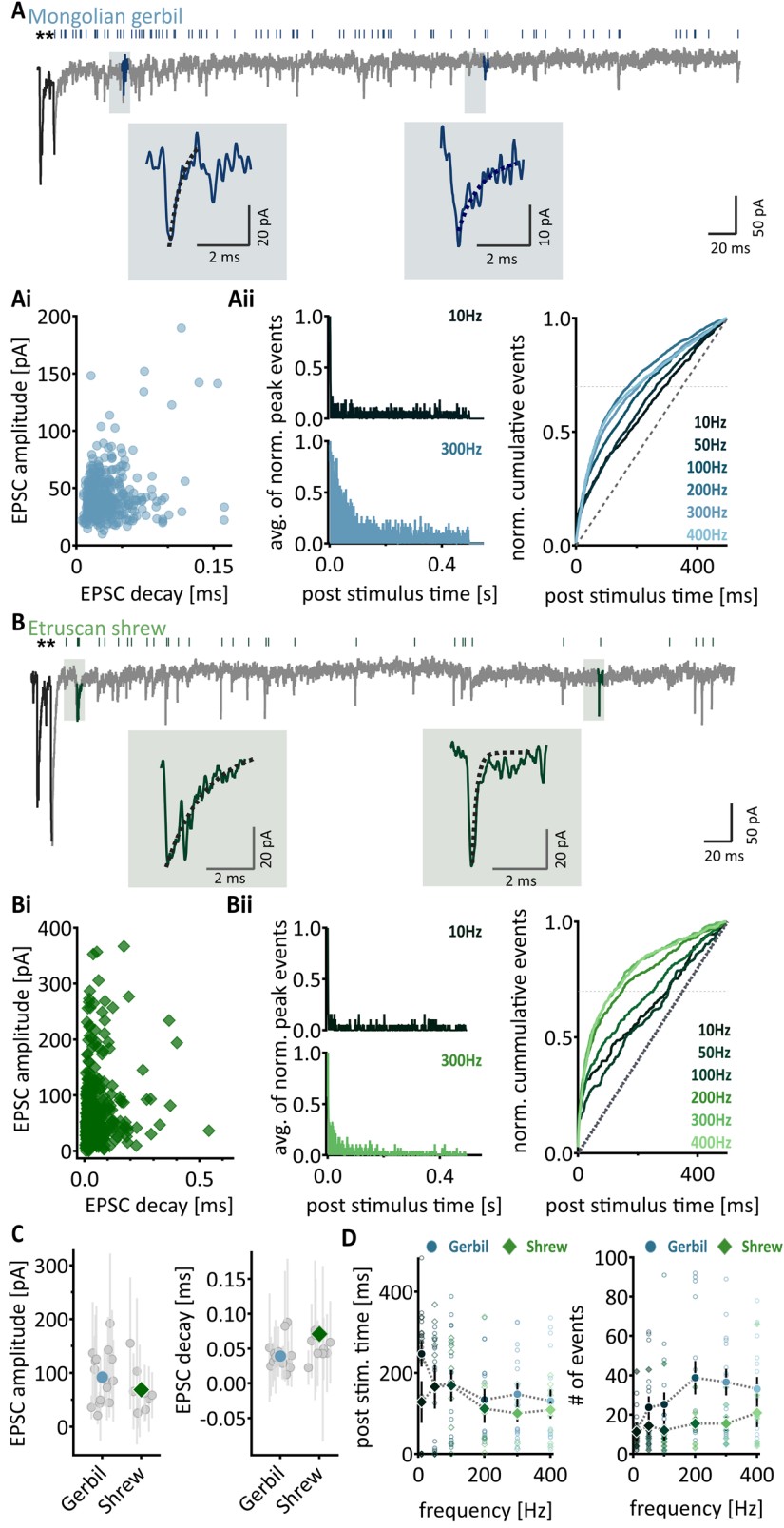

gerbils and Etruscan shrew, individual EPSC events of delayed release were isolated and their amplitude was determined by finding the minimum peak. From that minimum, we further fitted a biexponential curve to calculate individual EPSC decay times. The extracted events from one recording were pooled, their average and standard deviation were determined. The large standard deviation underlines the large heterogeneity of asynchronous

EPSCs within a recording (Fig. 2A, B). The averages of the extracted events showed that delayed released EPSCS were similar in amplitude (p = 0.2136) and tended to be faster in decay (*p* = 0.0568) in gerbils compared to Etruscan shrew (Fig. 2C).

In both species, the number and temporal occurrence of delayed release events were frequency-dependent (Fig. 2Aii and Bii). High stimulation

frequencies led to a slower decay of delayed release component compared to low stimulation frequencies, as seen in the normalized cumulative EPSC distribution by the steeper increase in event occurrence (Fig. 2Aii and Bii). To capture the differences in the time course, the post-stimulation time when 70% of the normalized delayed release occurred was measured (Fig. 2Aii and Bii). The time of 70% occurrence of delayed release decreased in both species from low to high stimulation frequencies (Fig. 2D), indicating a relative increase in the prolongation of delayed release. The frequency dependence was stronger in gerbils where at low stimulation frequencies the delayed release decayed more rapidly (Fig. 2D). Moreover, the number of detected delayed EPSCs appeared larger in gerbil compared to shrew at least for some frequencies (Fig. 2D; 10 Hz: $p = 0.6236$; 50 Hz: 0.4754; 100 Hz: $p = 0.0381$; 200 Hz: $p = 0.1952$; 300 Hz: $p = 0.0404$; 400 Hz: $p = 0.2071$) and showed a stronger frequency-dependent increase (Fig. 2D, Gerbil: Friedman $p < 0.0001$; 10 Hz vs. 200 Hz: $p < 0.0001$; 10 Hz vs. 300 Hz: $p < 0.0001$; 10 Hz vs. 400 Hz: $p = 0.0009$; 50 Hz vs. 200 Hz: $p = 0.0373$; 50 Hz vs. 300 Hz 0.0095; Shrew: Friedman $p = 0.5526$).

Because the calyx of Held and the non-calyceal inputs display dissimilar STP forms, we reasoned that the vesicular refilling dynamics and calcium dependence of the STP phenotype might differ between both synaptic input types. Since the vesicle dynamics of gerbil calyx of Held synapses have been elucidated[30,39] and the calcium-dependent shift from STD to facilitation of the calyx synapse is well documented, we focused on the non-calyceal inputs in gerbils to allow for a direct comparison. To determine calcium dependency of STP and the refilling time course, vesicle pool depletion was enhanced by an elevation in extracellular calcium concentration, which is considered to increase the release probability and thus benefits pool depletion. Therefore, afferent fiber stimulation of 30 pulses at 200 Hz was performed in a separate group of inputs ($n = 14$) in elevated extracellular calcium concentrations of 2.5 mM. Moreover, to account for the asynchronous release during the stimulation trains the quantification of vesicular dynamics was based on the EPSC charge. Under these conditions, the non-calyceal inputs showed STD instead of STF, arguing for the proposed increase in release probability (Fig. 3A). The normalized charge profile corroborates this depressing STP profile and demonstrates the difference to the facilitating EPSC charge profile in 1.2 mM external calcium concentration (Fig. 3A). According to the proposed increase in release probability, the charge of the first EPSC was increased between both sample groups (Fig. 3B; $p = 0.0058$). To assess the recovery of the apparent readily releasable vesicles, two train stimulations were paired with different inter-stimulus intervals ranging from 61 ms to 32 s and the charge of the EPSC responses during the trains were extracted (Fig. 3C). The recovery of the normalized EPSC charge of the full train could be approximated by a bi-exponential fit with a fast (80 ms) and a slow time constant (3.6 s; Fig. 3D). These recovery time constants are similar to the pool recovery of the calyx of Held in gerbils[30], despite the fact that their output and release probability under physiological conditions strongly differed.

Under physiological conditions, the calyx of Held input in the MNTB is spontaneously active[21,39,40]. To mimic the ongoing activity in vitro and investigate the STP dynamics under these conditions, we analyzed the STP profiles for non-calyceal and calyx of Held inputs in gerbils after a pre-conditioning of 20 Hz for 21 s under 1.2 mM external calcium concentrations (Fig. 4A, B). For non-calyceal inputs, the steady state EPSC size after 21 s pre-conditioning leveled at the size of the initial EPSCs (Fig. 4A). For the calyx of Held inputs the steady state EPSC size after 21 s pre-conditioning was depressed to a fraction of about 0.55 of the first EPSC (Fig. 4B). After the 21-second pre-conditioning, high-frequency trains of 30 pulses at frequencies between 100 and 400 Hz were applied (Fig. 4A, B). Similar to the unconditioned paradigm, the STP of non-calyceal and calyx of Held inputs showed frequency-dependent facilitation and depression, respectively (Fig. 4A–E; NC: ANOVA $p = 0.3811$ Calyx: ANOVA: $p < 0.0001$; 100 Hz vs. 200 Hz: $p < 0.0001$; 100 Hz vs. 200 Hz: $p < 0.0001$; 100 Hz vs. 300 Hz: $p < 0.0001$; 100 Hz vs. 400 Hz: $p < 0.0001$; 200 Hz vs. 300 Hz: $p = 0.006$; 200 Hz vs. 400 Hz: $p = 0.0006$). For the calyx of Held, due to the partial pre-depression, the short-term dependent decrease of absolute EPSC size

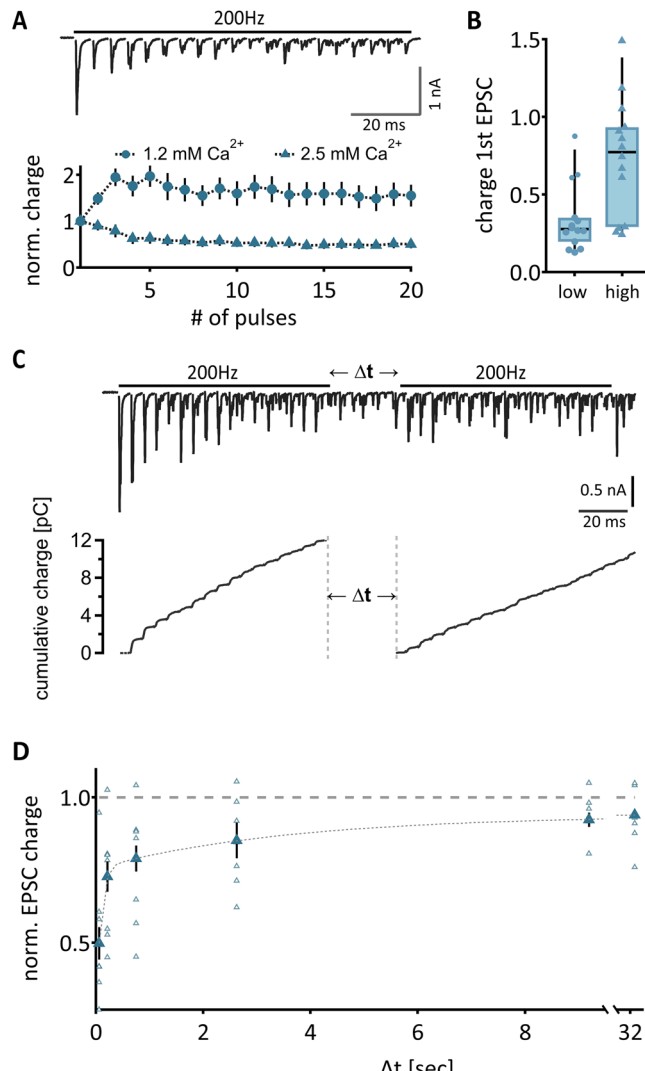

**Fig. 3 | Calcium dependency and recovery dynamics of non-calyceal input size in Mongolian Gerbil. A** Top: Exemplary EPSC response to a stimulation train of 200 Hz in elevated calcium (2.5 mM). The black line indicates duration of stimulation train. Bottom: normalized charge of EPSC in response to the train stimulation. Circles show data recorded at physiological calcium concentration. Triangles show evoked EPSCs in response to elevated calcium concentration. **B** Comparison of the charge of the 1st evoked EPSC in low and high external calcium concentration. Blue symbols show the cell average, and black symbols show the median ± quartile. **C** To elucidate the recovery dynamics of non-calyceal inputs, a 20-pulse stimulation, followed by a second 20-pulse stimulation train with increasing inter-stimulus interval (Δt) was applied. Top shows exemplary EPSC train to this stimulation. Bottom: cumulative EPSC charge to estimate the recovery time course. Dotted vertical lines highlight the inter-stimulus interval. **D** Recovery dynamics of non-calyceal inputs. The dashed line represents 100% recovery. Triangular symbols show normalized EPSC charge (mean ± SEM). Single cell data ($n = 14$). depicted in open symbols.

induced by the high-frequency train was less compared to the initial first EPSC amplitude in the entire stimulation train. Thus, the extracted STP after pre-conditioning normalized to the steady-state level of the pre-conditioning appeared slightly lower compared to the unconditioned case (Figs. 4D and 1G). For calyx of Held inputs, the pulse number that generated the largest EPSC in the high-frequency train after conditioning was again either the first or the second one (Fig. 4E). For non-calyceal inputs, we observed facilitation during high-frequency trains after conditioning, but the frequency dependence of this facilitation was altered. Similar to unconditioned recordings, the maximum EPSC occurred later in the

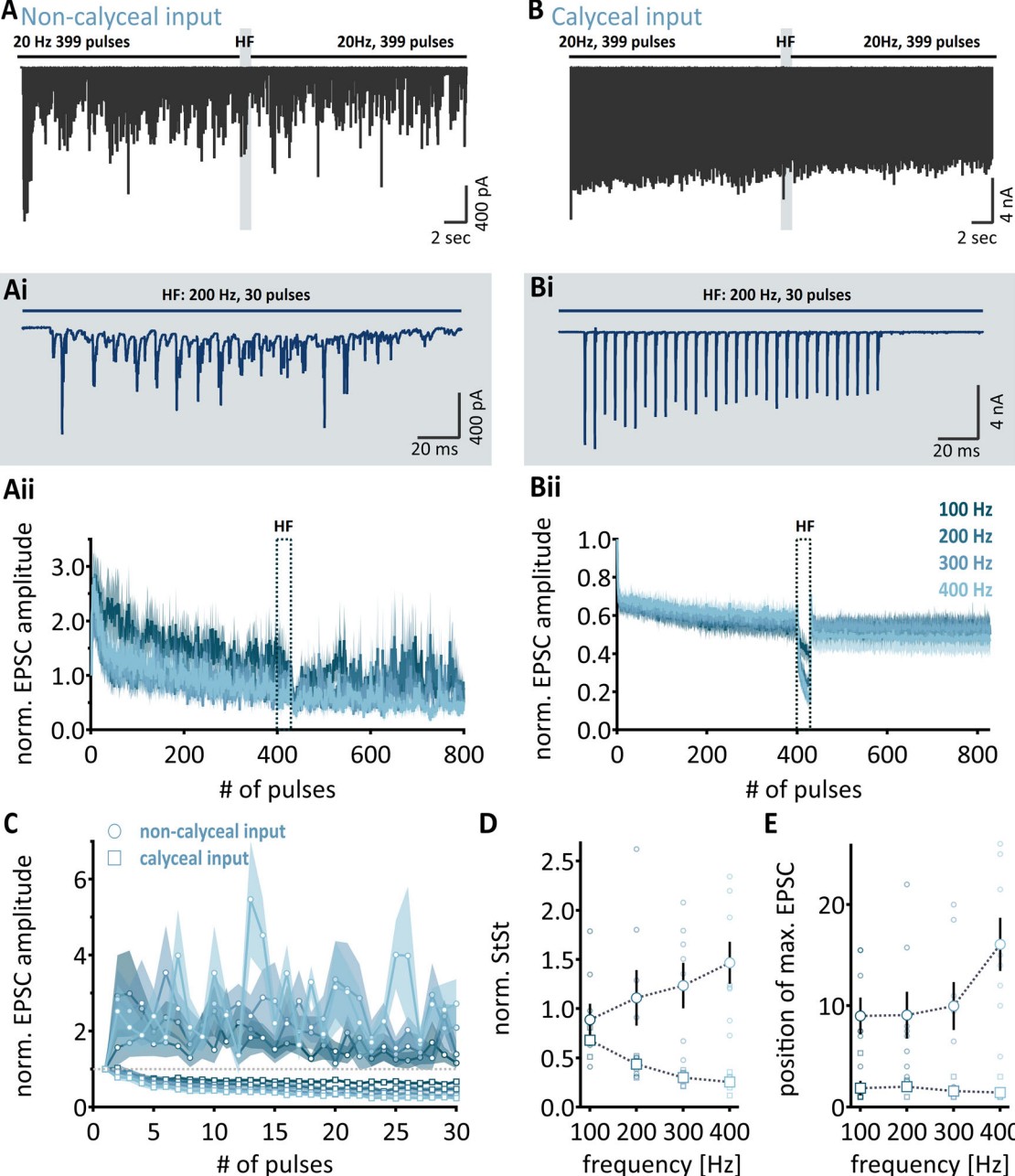

**Fig. 4 | Characterization of short-term plasticity under the influence of synaptic pre-conditioning in gerbil. A** Exemplary recording of non-calyceal synaptic input response to a conditioning protocol. After a 20 Hz pre-conditioning stimulation for 399 pulses a high-frequency train (200 Hz) was applied and finally followed by another 20 Hz pulse sequence for 399 pulses. **Ai** shows an inset of the high-frequency train. **Aii** Normalized EPSC amplitude plotted against each stimulation pulse. Dotted frame highlights high-frequency stimulation. **B** to **Bii** same as (**A–Aii**) but for calyceal inputs. **C** Normalized EPSC amplitude of calyceal (bottom) and non-calyceal input (top) during stimulation pulses at high-frequency trains (100 Hz, 200 Hz, 300 Hz, 400 Hz). Circles show non-calyceal input, and squares depict calyx of Held. **D** and **E** Average ( ± SEM) normalized steady state and position of maximum EPSC occurrence per frequency. Circles show non-calyceal input, and squares depict calyx of Held inputs, n = 10. Single cell data is shown in small symbols.

ongoing stimulation for non-calyceal inputs compared to calyx of Held inputs (100 Hz: p = 0.0033; 200 Hz: *p* = 0.0219; 300 Hz: *p* = 0.0023; 400 Hz: *p* = 0.0007). After conditioning, higher stimulation frequencies above 200 Hz induced robust steady-state facilitation, while frequencies below 200 Hz either leveled the steady state to the pre-conditioned level or evoked a slight steady-state depression. This is contrary to the responses of the unconditioned stimulation paradigm, where high-frequency stimulations above 200 Hz led to a steady state of one or slightly below, while frequencies below 200 Hz evoked robust steady state facilitation (Figs. 4D and 2G). A similar conversion was observed for the pulse that generates the largest EPSC within the high-frequency pulse train (Figs. 4E and 2I). Taken

together, under near physiological conditions the non-calyceal inputs are capable of sustaining an increase in EPSC response during high-frequency stimulations and therefore might be able to compensate for the depression of calyx of Held inputs.

### Functional significance of non-calyceal inputs to MNTB output generation

To functionally corroborate that non-calyceal inputs influence calyx of Held-mediated high-frequency output generation in MNTB neurons, we employed dynamic clamp recordings. Toward this aim, we created conductance templates based on the average form of unconditioned (Fig. 5A)

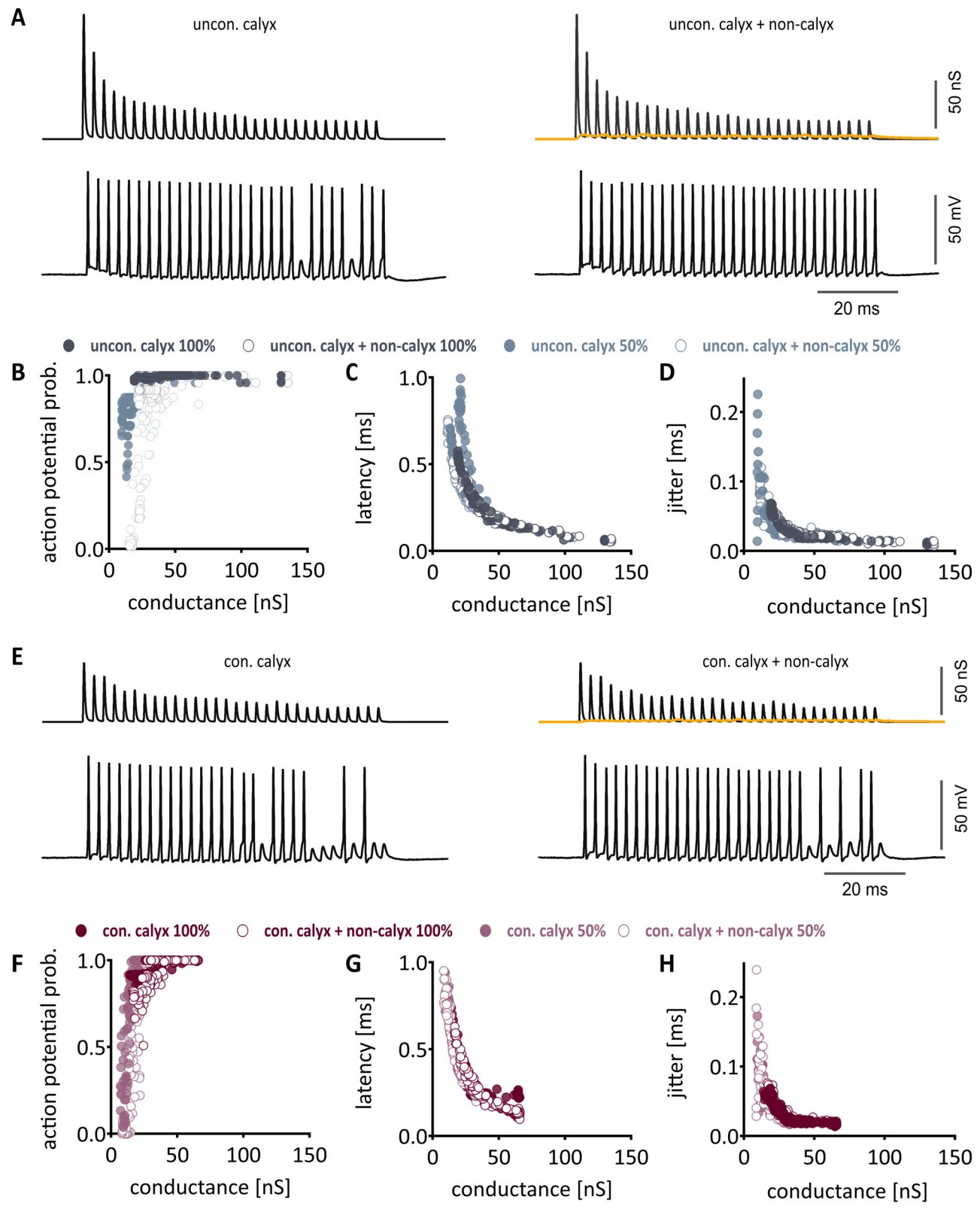

and pre-conditioned short-term behavior (Fig. 5E) of calyx and non-calyx inputs. These templates were either applied alone or combined in the following succession: unconditioned non-calyx, pre-conditioned non-calyx, unconditioned calyx plus non-calyx, pre-conditioned calyx plus non-calyx, unconditioned calyx, pre-conditioned calyx. Moreover, the templates were scaled either to match the recorded average EPSC size (100%) or to 50% of

the recorded average EPSC size. This reduction in conductance amplitude aims to mimic the EPSC size for neurons that have higher spontaneous input frequencies [21,39,40].

Conductance of non-calyceal inputs alone typically did not produce a supra-threshold output and were not considered further for the analysis of suprathreshold output generation. We assessed whether the total, summed

**Fig. 5 | Functional implication of additional non-calyceal conductance in gerbil.**
**A** Exemplary conductance template of calyceal (black) and additional non-calyceal
(yellow) input at 100% intensity without preceding low-frequency stimulation
(unconditioned). Below, the corresponding postsynaptic cell response is shown. On
the left, only calyceal templates and the response to those are shown; the right shows
the combined conductance of calyx of Held input and additional non-calyceal
inputs. **B–D** From left to right: Average action potential probability, latency, and
jitter as a function of applied unconditioned conductance (blue) in all cells and
shown independently of stimulation frequency. Calyceal only stimulation is

depicted by filled circles, combined conductance of calyx of Held and non-calyceal
inputs is displayed as unfilled circles. Intensity of stimulation is depicted by the hue
of color, where dark blue data points were acquired during 100% stimulation
intensity, and light blue data points during 50% stimulation intensity. **E** Similar to
(**A**), but with preceding low-frequency stimulation (conditioned). **F–H** Similar to
(**B–D**) but with conditioned conductance application (magenta). Respectively data
points acquired during 100% stimulation are shown in dark magenta and data point
acquired at 50% stimulation in light magenta. Data are presented as average ±
SEM, $n = 16$.

---

EPSG amplitude at the position of each pulse in the stimulation train is the
main determinant of the generated output, or if the integration of calyx and
non-calyx inputs leads to a specific, non-linear output generation. We linked
the success rate, latency and jitter for each condition and template scaling
according to the applied, total EPSG amplitude. Success rate (Fig. 5B, F),
latency (Fig. 5C, G) and jitter (Fig. 5D, H) showed a dependency on the total
conductance amplitude, irrespective of conditioning and combining of
templates. The dependence of the conductance on action potential gen-
eration is steep and a small difference below 30 nS of max conductance can
lead to a high variability of output generation. Together, this corroborates
previous findings in MNTB and VNLL[28,41] showing that synaptic con-
ductances are the main drives and add at the soma linearly in respect to the
generating suprathreshold output.

It is functionally relevant to understand to what extent excitatory non-
calyceal inputs are able to temporally modulate the output generation of
MNTB neurons driven by the calyx of Held. To assess this integration effect
on temporal output generation, we paired afferent fiber stimulation trains of
50 non-calyceal inputs with trains of 50 calyx of Held specific conductance
templates. Train stimulation of non-caclyceal input generated the pre-
viously described EPSCs. Those evoked EPSCs translated into EPSPs, which
generated a slight plateau depolarization, caused by the temporal overlap of
two subsequent EPSPs (Fig. 6A). Pairing of these voltage responses with
supra-threshold calyx of Held EPSGs shifted the timing of action potentials,
(Fig. 6B). To quantify this temporal shift, the latency between peak calyx of
Held EPSG and the peak of the action potential was determined for train
frequencies of 100, 200, 300, and 400 Hz for each pulse in the train. The calyx
of Held EPSGs were scaled in size to match the steady state of these sti-
mulation frequencies. The latency difference between paired stimulation of
non-calyceal with simulated calyx of Held inputs increased in a frequency-
dependent manner from 5.66 µs (1.4–10.8 µs) at 100 Hz to 50.55 µs
(19.8–93.3 µs) at 400 Hz ($n = 11$; Fig. 6C). Thus, non-calyceal inputs have
significant, frequency-dependent influence on the output timing of the calyx
of Held (Friedman, $p = 0.0002$, 100 Hz vs 300 Hz, $p = 0.0178$; 100 Hz vs
400 Hz, $p = 0.0001$; 200 Hz vs 400 Hz, $p = 0.0494$). Because action potential
generation is voltage-dependent, the latencies obtained by pairing non-
calyceal inputs with calyx of Held EPSGs were correlated with the corre-
sponding depolarization, induced by the average non-calyx of Held EPSPs.
This analysis was performed for each supra-threshold event throughout the
train in each recorded neuron (Fig. 6D). Like before, latency of action
potentials depended on applied stimulation frequency. Furthermore,
latency also depended on the depolarization induced by the non-calyceal
inputs. Larger depolarizations reduced the latency for each stimulation
frequency tested (Spearman R coefficient, 100 Hz, R = -0.551, $p < 0.0001$;
200 Hz, R = -0.451, $p = 0.001$; 300 Hz, R = -0.377, $p = 0.0069$; 400 Hz, $p = -0.655$, $p < 0.0001$; Fig. 6D).

To assess the full dynamic range of sub-threshold evoked shifts in
supra-threshold output timing of non-calyceal inputs, simplified con-
ductance templates for non-calyceal inputs were paired with calyx of Held-
like EPSGs. The calyx of Held-generated output was induced by a con-
ductance amplitude that mimicked steady-state EPSC sizes ($G_e$) for fre-
quencies between 100 and 400 Hz. The non-calyceal input was simplified by
a non-fluctuating conductance with rapid rise, constant steady-state level
and a slow decay to match the STP kinetics of these inputs (Fig. 6E). Finally,
this modulatory input was scaled between 0 and 6 nS for excitatory inputs

(modulation $G_e$) to account for possible multiple non-calyceal inputs to a
given neuron. Because MNTB neurons also receive facilitating inhibitory
inputs with asynchronous release[42], we extended this protocol for inhibitory
conductances (modulation $G_i$) by inverting the non-calyceal conductance
ranging from 0 up to 15 nS (Fig. 6F). The conductance of 0 was applied for
both excitatory and inhibitory modulation and hence responses overlapped.
The modulatory conductances were applied with and without the calyx of
Held conductance template to obtain the change in membrane potential
(Fig. 6E, F).

Calyx of Held conductances (pre-conditioned: 100 Hz: 48.8 nS;
200 Hz: 33.9 nS; 300 Hz: 24.5 nS; 400 Hz: 18.5 nS; unconditioned: 100 Hz:
63.2 nS; 200 Hz: 50.5 nS; 300 Hz: 34.6 nS; 400 Hz: 24.8 nS) generated action
potentials without failures throughout the stimulation train, irrespectively
whether it was combined with modulatory $G_e$ or $G_i$ (Fig. 6E, F). From the
single applied modulatory conductance, the induced change in membrane
potential was obtained without interference from action potential genera-
tion. The change in membrane potential depended on the amplitude of the
modulatory conductance and reached for the strongest $G_e$
$13.19 \pm 0.125$ mV and for the strongest $G_i$ -$7.43 \pm 0.154$ mV ($n = 10$;
Fig. 6G). The major finding of this experiment, however, was that the timing
of the calyx conductance-generated action potential was altered. Mod-
ulatory $G_e$ led to reduced latency, while $G_i$ increased the average latency of
the action potentials during co-application of both conductances (Fig. 6E,
F), compared to the preceding action potentials evoked by only the calyx of
Held conductance. Because of the modulatory conductance applied alone,
we were able to link the shift in action potential timing to the change in
membrane potential. The shift in latency depended on both the frequency of
the calyx conductance and the reached membrane potential induced by
different modulatory conductance amplitudes (Fig. 6H). The maximal
difference in temporal shift of 244.34 µs was between the depolarization of
12.06 mV and hyperpolarization of -6.68 mV, both induced by combined
conductance the calyx of Held simulation of equivalent to 400 Hz and the
summed excitatory and inhibitory conductance respectively. A similar
temporal shift is apparent within all stimulated frequencies (Friedman:
100 Hz: $p < 0.0001$; 200 Hz: $p < 0.0001$; 300 Hz: $p < 0.0001$; 400 Hz
$p < 0.0001$). The precision, measured as the average jitter during a co-
applied conductance train was also affected at calyx of Held stimulation
frequencies of 100 and 400 Hz and depended also on the leveled membrane
potential (100 Hz: $p = 0.02$; 200 Hz: $p = 0.0005$; 300 Hz: $p = 0.0159$; 400 Hz:
$p < 0.0001$; Fig. 6I). Taken together, integration of modulatory conductances
mimicking synaptic inputs, together with the calyx of Held input at the
MNTB principal neuron adjusts the timing of the supra-threshold output
generation and therefore adds to the functions beyond a simple relay
mechanism.

## Discussion

We show that the integration of opposing forms of STP, originating from
different input sources, modulates the temporal precision and success of
supra-threshold output generation. In MNTB neurons, this integration is
given by the depressing calyx of Held and the facilitating non-calyceal
inputs. The supra-threshold output of MNTB neurons is computed by the
summation of these distinct input conductances. Moreover, non-calyceal
inputs gate the temporal precision of calyx of Held generated output. This
supra-threshold output modification is based on facilitation and

**Fig. 6 | Non-calyceal conductance can shift latency and jitter in gerbil. A** Top: Example of EPSC response to fiber-stimulation of non-calyceal inputs. Dotted line depicts the duration of stimulation. Bottom: Corresponding EPSP response of the same stimulated non-calyceal input. **B** Top: Example of applied calyceal conductance template. X labels the time point of fiber-stimulation of non-calyceal inputs. Bottom: Action potential response. Gray traces and arrows mark the shift in latency when both conductance and fiber stimulations are applied simultaneously. **C** Average change in latency induced by simultaneous application of conductance template and stimulation of non-calyceal inputs in dependence of stimulation frequency. Data are shown as median ± 1 and 3rd quartile, $n = 11$. Single cell data is shown with open symbols. **D** Averaged latencies of each action-potential during stimulation (50 pulses) at all four stimulation frequencies in dependence of total change in membrane potential ($E_{mem}$). Lines display correlation. Data are shown as mean ± SEM, $n = 7$. **E** Top: Exemplary conductance template of calyceal steady state conductance. The inset shows one EPSG of the calyx of Held template. Below, an exemplary modulatory, excitatory non-calyceal conductance template is depicted. The modulatory template was scaled in peak conductance to mimic an increase of synaptic input size. The lower panel shows the cell response to the applied conductance. The modulatory conductance alone was not able to evoke any action potentials. The inset shows a shift in action potential timing when the additional modulatory conductance (gray) was applied. **F** Same as (**E**) but with inhibitory modulation template of non-calyceal inputs. **G** Dependence of the difference in membrane potential on applied modulating conductances, $n = 10$. Single cell data is shown with open symbols. **H** and **I** Average latency and jitter depending on the change in membrane potential. Different hues of blue indicate different applied simulated calyx of Held frequencies. Data are shown as mean ± SEM. Single cell data is shown with open symbols. The dashed line marks no change in membrane potential (zero line), $n = 18$ for 100 Hz; $n = 17$ for 200 Hz; $n = 8$ for 300 Hz; $n = 8$ for 400 Hz.

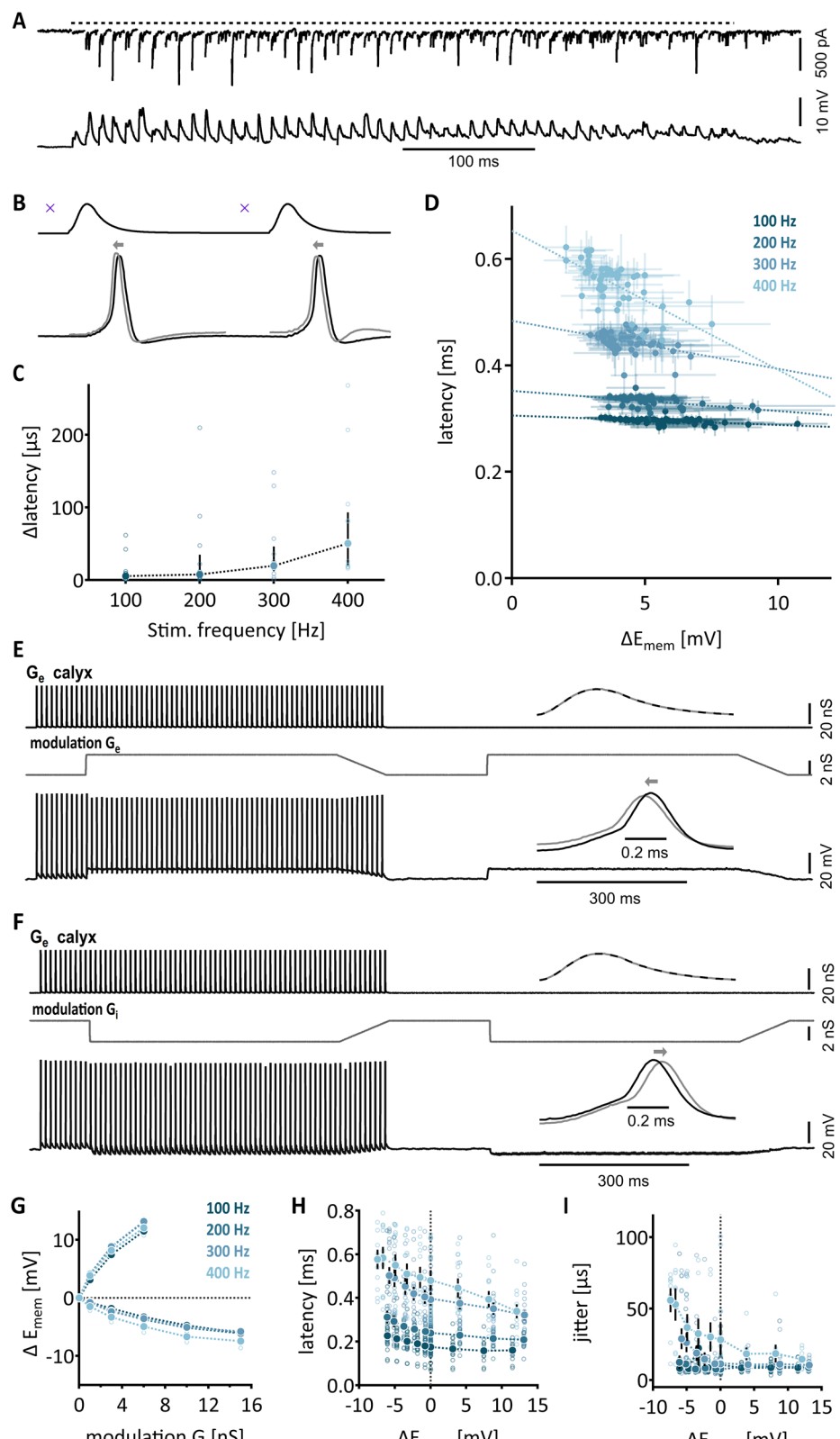

asynchronous release of the non-calyceal inputs. Mechanistically, synaptic integration drives a subthreshold shift in membrane potential, adjusting the output timing of the MNTB neurons, thus intervening on multiple levels with sound processing.

Non-calyceal and calyx of Held inputs are readily distinguished by the EPSC waveform and STP. The non-calyceal inputs generate small

fluctuating EPSCs[34] with a large heterogeneity in decay time constants, while calyx of Held synapses evoke large and fast EPSCs[28,29]. At near physiological conditions, we observed the documented STD in the calyx of Held[28,30] and robust facilitation in non-calyceal inputs. This facilitation was accompanied by a breakdown in phasic release response, as the occurrence of jittered asynchronous EPSCs increased during the stimulation train. For both input

types, the STP was use- and frequency-dependent. During ongoing activity, for non-calyceal inputs, the increase in asynchronous release together with facilitation, and for the calyx of Held inputs the stronger depression, indicate a change in temporal precision of transmitter release. Thus, during the high-frequency activation, both inputs individually are bound to lose temporal precision. For the calyx of Held input, this effect is even stronger after pre-conditioning the inputs with prolonged ongoing activity. A stronger depression during such ongoing stimulations has been reported before for the calyx of Held and other auditory synapses[39,43]. The effect of this pre-conditioning on the calyx of Held synapse is that it decreases absolute EPSC amplitudes during STD even stronger. However, the relative depression between unconditioned and the post pre-conditioned situation remains similar. For the non-calyceal inputs, the pre-conditioning altered the absolute EPSC size during the following high-frequency stimulation only moderately. However, the facilitation became more long-lasting and its frequency-dependence changed. Compared to the unconditioned situation, high stimulation frequencies led to stronger facilitation, while low frequencies resulted in lower facilitation. Whether this change in frequency-dependence of the non-calyceal facilitation is driven by the accumulation of residual calcium, an activity-dependent change in release or recovery rates during pre-conditioning remains elusive. Taken together, under physiological conditions the temporal precision of transmitter release of both input types is lowered compared to stimulations without prior use-dependence.

Besides the large physiological differences between the non-calyceal and the calyx of Held inputs, there seems to be at least one similarity. An increase in the external calcium concentration led to a depression in the non-calyceal synapse that induced an apparent depletion of the releasable pool. The apparent readily releasable pool in non-calyceal inputs recovered biexponentially with time constants similar to those described for calyx of Held synapses[30]. Thus, we suggest that under physiological conditions, where a short bout of facilitation might apparently depress the releasable pool, the rapid refilling promotes the observed sustained ongoing release at non-calyceal synapses.

Next to the difference in STP, the temporal precision of transmitter release differs between non-calyceal and calyx of Held synapses. While the calyx of Held maintains phasic release events during stimulation trains, non-calyceal inputs show substantial asynchronous release, which transitions into a stimulation-frequency dependent delayed release. For non-calyceal synapses, the amount of delayed release is highly heterogeneous. This heterogeneity possibly originates from the different sources of input fibers. Potential sources of these excitatory non-calyceal fibers are collaterals of calyx of Held inputs and fibers from the cochlear nucleus[33], as well as descending fibers from the semilunar nucleus close to the lateral lemniscus[44]. The substantial asynchronous and delayed release from non-calyceal synapses indicates differences in calcium handling or variation in calcium sensor expression compared to the calyx of Held. For example, an activity-dependent increase in local calcium transients to support facilitation and release of vesicles with larger distance to calcium channels could be mediated by a low amount or rapidly saturating calcium buffers[4,5]. Moreover, as Synaptotagmin 7 has been implicated in asynchronous transmitter release[45–47] a differential expression could be a major difference between non-calyceal and calyx of Held synapses. Functionally, the frequency-dependent buildup of asynchronous release together with the facilitation provides MNTB neurons with a frequency-adjusted increased DC current leading to ongoing sub-threshold depolarization. The ongoing conductance and depolarization are thereby integrated together with the calyx of Held conductance and at least partially counteract the STD of the calyx of Held.

The calyx of Held - MNTB relay is well known for its temporal accuracy[21,22,48]. Yet, at this synapse, temporal precision is lost during ongoing activity[48]. Our analysis agrees with recent results[28] that synaptic depression can be a key component of this loss in precision, because the sum of the absolute synaptic conductance determines the timing and precision in the MNTB. Since the facilitating non-calyceal inputs can be simplified as a DC current, they will partially compensate for the loss in accuracy induced by the STD of the calyx of Held. Moreover, our conductance-clamp recordings

allowed us to determine the contribution of the different inputs to the temporal accuracy. By adjusting the size of calyx of Held EPSGs to mimic in vivo-like timing of ongoing activity[22,48], we can estimate a potential the shift in precision of action potentials by the non-calyceal input conductance. Our findings show that these inputs shift the timing of MNTB supra-threshold output over the relevant range for auditory processing times (30 – 150 μs), as this range matches the functional ITD range in gerbils. Because the MNTB feeds forward its inhibition to the binaural detectors, this temporal shift should be capable to functionally modify binaural detection and hence sound source representation. Since non-calyceal inputs might arise from higher centers by descending tracts, their powerful modulatory influence might be a feedback loop to adjust the detection and representation of sound sources similar to local feedback systems described for the medial and lateral superior olive[19,20]. However, to translate our in vitro findings, acquired in an experimental environment of limited modulation, background and no sensory stimulation, to the in vivo situation remains limited and speculative.

The integration of synaptic conductance of non-calyceal and calyx of Held inputs determines the output function of MNTB neurons. The integration, especially of non-calyceal conductances, shifts the membrane potential of MNTB neurons. Since next to excitatory non-calyceal inputs, also inhibitory synapses to MNTB neurons that facilitate and produce asynchronous release[32,42] exist, the shift in membrane potential might as well not only be depolarizing, but also hyperpolarizing. Thereby, the shift in membrane potential defines the shift in temporal accuracy by acceleration and deceleration to generate supra-threshold output in the range between 70 – 170 μs. Therefore, the synaptic sign adds to and enhances the temporal modulation based on changes in leak conductance[49]. The conductance-based integration of the asynchronous release of non-calyceal inputs suggests furthermore that their timing in respect to the accuracy of the calyx of Held might be less relevant. Overall, even in the relay station of the MNTB the balance between non-calyceal excitatory and inhibitory inputs adjusts sound processing.

Our comparative data support the general structure-function of the MNTB in mammals. In gerbil (Rodentia) and Etruscan shrew (Eulipotyphla) the synaptic phenotypes are the same despite 94 MYA of segregation (http://www.timetree.org/). Yet, as within rodents[50] and between distal mammalian species[28], also gerbils and Etruscan shrews show species-specific modifications of this general scenario. Compared to Etruscan shrew, the non-calyceal inputs of gerbils show slightly weaker facilitation, are less sustained and the delayed release kinetics show less frequency dependence. These differences might indicate a difference in global calcium homeostasis in the synaptic boutons matching the near absence of calcium binding proteins in Etruscan shrew[51]. The calyx of Held generates larger EPSCs and more depression in gerbils compared to Etruscan shrews. One explanation for the EPSC size difference might be a different initial release probability, in agreement with the larger STD in gerbils compared to Etruscan shrew. The molecular and cellular mechanisms for a possible different initial release probability are so far unclear. Functionally, the EPSC size reduction in Etruscan shrew might be adapted to match the so far unexplored post-synaptic properties of these overall smaller neurons. Alternatively, like bats, Etruscan shrews might be more adapted to ongoing sounds than gerbils[28] or gerbils and other rodents show a specific adaptation at this synaptic junction.

Taken together, different inputs to the same auditory neurons have different use-dependent synaptic filter functions that are integrated on the basis of their conductance to adjust the action potential generation to the appropriate rate and timing, for example relevant for representing sound sources in space.

## Materials and methods
### Animals and slice preparation
The data presented in this study were obtained from a total of 45 Mongolian gerbils (*Meriones unguiculatus*) and 12 Etruscan shrews (*Suncus etruscus*) of both sexes. Gerbils, based on Charles River hereditary background, and shrews were bred and kept in the animal facility of the Institute of Zoology.

Gerbils had access to food and water supply *ad libitum* and were kept under a 12 h light/dark cycle. Etruscan shrews were housed as described before[51]. The age of gerbils used in this study ranged from postnatal day (P)24 to P41 (30.5 ± 0.6 days), and for shrews from two to 18 months (7.6 ± 1.3 months). In both age groups hearing is fully established and no substantial age-related detrimental effects for hearing are expected. In our preliminary auditory brainstem response data age-related detrimental effects were mildly apparent in Etruscan shrew older 18 months compared to 6-12 months. While Mongolian Gerbils serve as a common model in hearing research, a comparative approach illustrates generality. Etruscan shrews are exquisite hunters that rely next to touch on hearing to detect their prey even at night and therefore require precise spatial information from the auditory system. In both gerbils and Etruscan shrew, the medial nucleus of the trapezoid body is well developed and contains large somatic synapses. MNTB mediated inhibition is relevant at least for sound source localization targeting MSO and LSO. In Etruscan shrew no clear structure MSO is detectable[51] consistent with their exclusive high frequency listening above 6 kHz. Thus, in comparison to gerbils, which hear between 0.5 and 50 kHz, Etruscan shrews are specialists for high frequency listening.

All animals were sacrificed in accordance with federal law and local regulations, and the local animal welfare officer approved the procedure under the local animal protocol (TiHo-T2021-4 and TiHo-T2024-3). Gerbils and shrews were deeply anesthetized by inhalation of isoflurane, decapitated and brains were rapidly removed in ice-cold preparation solution consisting of (mM): 93 NMDG, 93 HCl, 30 NaHCO$_3$, 1.2 NaH$_2$PO$_4$, 2.5 KCl, 25 glucose, 20 HEPES, 5 ascorbic acid, 3 myo-inositol, 3 sodium pyruvate, 10 MgCl$_2$, 2 CaCl$_2$ bubbled with 95% O$_2$ and 5% CO$_2$ to saturate the solution with oxygen and adjusted the pH to 7.4. Brains were transversally trimmed before slices with a thickness of 200 µm, containing the MNTB, were taken using a VT1200S vibratome (Leica). Slices were incubated for seven minutes in slice solution at 34 °C in a water bath and stored thereafter at room temperature in recording solution containing (mM) 125 NaCl, 25 NaHCO$_3$, 2.5 KCl, 25 glucose, 1.25 NaH$_2$PO$_4$, 1 MgCl$_2$, 1.2 CaCl$_2$, 0.4 ascorbic acid, 3 myo-inositol, 2 Na-pyruvate.

### In vitro electrophysiology

Brain slices were transferred to an upright microscope (Olympus BX51W1) equipped with a Retiga 2000DC camera or pco.edge 3.1 camera. Whole-cell recordings from visually identified neurons were acquired using an EPC 10 amplifier (HEKA Elektronik, Ludwigshafen, Germany) controlled by the Patchmaster software (HEKA Elektronik, Ludwigshafen, Germany). During recording, the slices were continuously perfused with recording solution supplemented with 1 µM strychnine to block inhibitory synaptic inputs, and the temperature was kept at physiological conditions ( ~ 34 °C ± 2 °C). Recording and stimulation pipettes were made from borosilicate glass capillaries (outside diameter: 1.5 mm, inside diameter: 0.86 mm, length: 100 mm, BioMedical Instruments, Zoellnitz, Germany) using a micropipette puller. The tip resistance varied between 3 to 6 MΩ. Access resistance was compensated to a residual of 3 MΩ in voltage-clamp mode. In dynamic-clamp recordings, the access resistance was bridge balanced to 100%, and no holding current was applied. Electrophysiological data were filtered at 3 kHz and acquired with a sample rate of 10 or 20 µs. Except for dynamic clamp data, no data have been corrected for liquid junction potential (LJP).

### STP Recordings

To characterize STP of non-calyceal inputs, recording pipettes were filled with intracellular solution containing (in mM): 145 K-gluconate, 15 HEPES, 4.5 KCl, 7 Na$_2$- Na$_2$-phosphocreatine, 2 Mg-ATP, 2 K$_2$ + -ATP, 0.3 Na$_2$ + -GTP, and 0.5 K-EGTA, 0.05 Alexa 568. To find a synaptic input to the patched neuron, a second stimulation electrode was brought down onto the slice. To identify also non-calyceal synaptic inputs during voltage-clamp recordings (holding potential: -60 mV), two 200 µs biphasic stimulations with a 3.1 ms inter-stimulus interval were applied with amplitudes up to 59 V. The double pulse stimulation was relevant as in some cases the non-

calyceal inputs would only respond to the second in the pair of stimuli. To circumvent the difficulty that non-calyceal inputs would not elicit an EPSC during the first stimulation pulse but do in subsequent pulses within the train, and no notable difference in EPSC amplitude or decay between maximum and threshold stimulation was observed, larger voltage stimulations were typically selected over the threshold stimulation for data acquisition. When a non-calyceal input was detected, train stimulation protocols consisting of 30 pulses in varying frequencies (10, 50, 100, 200, 300, and 400 Hz) were applied. During stimulation trains, cells were held at -70 mV. For fiber-stimulation of the calyceal of Held input, the intracellular solution was changed to a cesium-based solution to enable optimized clamping conditions due to blockage of potassium channels. This solution contained (in mM): 130 Cs-gluconate, 10 HEPES, 5 Cs-EGTA, 5 Na$_2$-Phosphocreatine, 4 Mg-ATP, 0.3 Na$_2$-GTP, 0.05 Alexa 568. Calyx of Held inputs of 10 and 50 Hz stimulation frequencies were not recorded. Additionally, train-stimulation protocols were expanded by 399 pulses at 20 Hz both prior and following the high-frequency train to mimic in-vivo like condition where constant background activity has been reported[21,30,40]. STP recordings with prior simulated in vivo like activity are regarded as conditioned responses, while STP recordings without pre-conditioning are referred to as unconditioned STP data.

### Recovery from depletion

To elucidate the recovery dynamics of non-calyceal input, the extracellular Ca$^{2+}$ concentration was elevated using a high calcium recording solution containing (in mM): 125 NaCl, 25 NaHCO$_3$, 2.5 KCl, 1.25 NaH$_2$PO$_4$, 0.5 MgCl$_2$, 2.5 CaCl$_2$, 25 glucose, 0.4 ascorbic acid, 3 myo-inositol, and 2 Na-pyruvate to initially deplete the vesicle pool. The intracellular solution was similar to that used during calyx stimulation (see above). Two stimulation trains, each consisting of 20 pulses at 200 Hz, were applied. The inter-stimulus interval (0.2 ms, 5.2 ms, 17.7 ms, 61.5 ms, 0.21 s, 0.75 s, 2.6 s, 9.1 s, and 32.2 s) between both stimulations was increased with each sweep to gain insight into the recovery of the present vesicle pool. Recordings were conducted in the presence of strychnine (1 µM), R-CPP (10 µM), and SR-97731 (10 µM).

### Dynamic clamp recordings

To investigate the functional implications of non-calyceal inputs to MNTB neurons, we utilized dynamic-clamp recordings, as previously introduced by Yang, Adowski, Ramamurthy, Neef and Xu-Friedman[52]. To record action potentials, the potassium-based intracellular solution was used, resulting in an LJP of 16 mV[53], which was corrected online. The sample rate of the recordings was 20 µs.

Previously recorded EPSCs of calyceal and non-calyceal inputs were converted into conductance templates while considering the STP dynamics. This led to 30-pulse excitatory postsynaptic conductance (EPSG) templates of either calyceal or non-calyceal origin at four different frequencies (100 Hz, 200 Hz, 300 Hz, 400 Hz) in either unconditioned or pre-conditioned states. These templates were then either applied alone or combined with a 2 s break, resulting in the overall template design: unconditioned EPSG$_{nc}$, pre-conditioned EPSG$_{nc}$, unconditioned EPSG$_{nc+c}$, pre-conditioned EPSG$_{nc+c}$, unconditioned EPSG$_c$, and pre-conditioned EPSG$_c$. Additionally, to simulate a more steady-state-like condition, the aforementioned conductance templates were scaled down to 50% of the original conductance, and recordings were conducted.

It can be assumed that not only one but multiple non-calyceal inputs innervate MNTB neurons, including not only excitatory but also inhibitory fibers[32,33,44]. To further investigate this phenomenon, a second dynamic clamp experiment has been designed. Therefore, a steady-state calyx conductance template was created consisting of 30 EPSGs with frequencies of either 100 Hz, 200 Hz, 300 Hz, or 400 Hz. Conductance templates of the non-calyceal input were modeled by simplification of the conductance to a non-fluctuating conductance. Here, we ensured implementation of a rapid rise, a constant steady-state level, and a slow decay to account for the STP kinetics of these inputs. Calyx and non-calyceal templates were co-applied,

whilst the non-calyceal input was either scaled (to mimic more synaptic inputs) or inverted (to mimic inhibitory synaptic input).

## Data analysis

Data were analyzed using custom-written macros in IGOR Pro 9 (Wavemetrics). Custom written code is made available as supplementary code 1-5. Statistics were conducted using GraphPad Prism. Data were tested for normal distribution using Shapiro-Wilk test. Data is presented in box plots or as average ± SEM. The significance level was set at 0.05. For multiple comparisons depending on whether the data were normally distributed or not, repeated measurements ANOVA with post-hoc Tukey's multiple comparison or Friedman test combined with post-hoc Dunn's multiple comparison test were used, respectively. For comparing two data groups, either unpaired T-Test or Mann-Whitney U-Test was chosen depending on normality distribution. All data presented in this manuscript and the accompanying statistics are provided in an Excel file in supplementary data.

## Reporting summary

Further information on research design is available in the Nature Portfolio Reporting Summary linked to this article.

## Data availability

All data supporting the findings of this study are accessible in the provided supplement; additional data are available upon reasonable request.

## Code availability

All custom written Igor functions (IgorPro 9) are available upon reasonable request.

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

## Acknowledgements

We thank Dr. Kathrin Wicke and Dr. Christina Pätz-Warncke for comments on the manuscript.

## Author contributions

L.C.-M.: data acquisition, analysis, writing and editing of manuscript; F.J.: data acquisition, analysis; N.K.: data analysis; F.F.: supervision, concept, writing and editing of the manuscript, funding.

## Funding

University of Veterinary Medicine Foundation Hannover and DFG FE789/12-1. Open Access funding enabled and organized by Projekt DEAL.

## Competing interests

The authors declare no competing interests.
