## [Transparent Peer Review file · Communications Biology]

Non-calyceal inputs gate the timing of calyx of Held evoked MNTB output

Corresponding Author: Professor Felix Felmy

Version 0:

Reviewer comments:

Reviewer #2

(Remarks to the Author)

Principal neurons in medial nucleus of the trapezoid body (MNTB) receive excitatory input via the Calyx of Held, a huge excitatory synapse that has served as an important model to study many aspects of synaptic function. The Calyx is built for speed and reliability, underlying its role in sound localisation. However, in addition to calyceal inputs, MNTB neurons also receive additional excitatory and inhibitory inputs across their soma and dendrites. Comparatively little is known about these inputs, and this is where the work of Console-Meyer et al. provides new information. The authors here report the properties of non-calyceal excitatory inputs, including their history-dependence, and propose their functional role may be to improve the reliability and speed up the response timing of MNTB neurons during calyceal activation.

To do this, the authors separately recorded calyx and non-calyx excitatory synaptic input. They then used dynamic clamp to mimic synaptic activation and measure the influence over neuronal output. Overall the study provides new information (particularly demonstrating facilitation of non-calyceal synaptic input). However, a key experiment appears to be missing – i.e. combining direct activation of both calyceal and non-calyceal inputs in the same neuron, and measuring action potential output. The work is statistically sound and repeatable. In general, I found the significance and justification to be lacking for some of the experiments. In its current form, the paper may not be accessible to a general biological readership. I outline these issues below:

1. Comparison between species

It would be helpful to justify the choice of species, and relevance of comparing gerbils and Etruscan shrews.

Brain slices of different maturity were used for the two species (adolescent gerbils and adult shrews). Is this a confounding or a deliberate part of the experimental design? Potentially this difference could account for differences in EPSC amplitude and overall synaptic facilitation between the two groups. While there are certainly notable differences in amplitude and dynamics of calyx and non-calyx EPSCs, it is not clear how seriously should we compare differences between gerbil and shrew data given the age differences. Since the shrew data is dropped after Figure 2, it is not clear to me how important is in to the overall paper.

Line 109 – ‘In both species, non-calyceal inputs exhibit strong STF in contrast to strong STD of the calyx...’

Currently this is not easy to determine from the data, especially in the gerbil (Fig 1A,1E). Perhaps this could be made more convincing by zooming in on the first few pulses of the stimulus train? In the gerbil, high stimulation frequencies seem to be subject to neither facilitation or depression.

2. Voltage clamp recordings of calyceal vs non-calyceal EPSCs

Is it problematic that a different internal solution was used to measure STP for calyceal vs non-calyceal input? Presumably recordings with the potassium-based intracellular solution will have higher noise. Also if non-calyceal input is arriving on the dendrite, could there be a space clamp issue, exacerbated by use of a potassium-based solution? Both possibilities could impair measure of evoked non-calyceal EPSCs.

3. Vesicle refilling dynamics

The significance of studying the vesicle refilling dynamics of non-calyceal synapses could be made more accessible. Was there a specific hypothesis here and what is the physiological relevance? Perhaps data from calyx synapses could be overlaid in some way on Figure 3D so the comparison can be made obvious, and support the claim that calyceal and non-calyceal inputs exhibit the same dynamics.

4. Functional significance demonstrated with dynamic clamp

Figure 5 – I have a hard time understanding this figure. Mainly, it is difficult to assess differences in responses to conditioned and non-conditions inputs – for example subpanels a and c look almost identical in main Panel A. At present, it is difficult to get a sense of how non-calyceal conductances contribute to the neuron output.

Line 293 – ‘Non-calyceal inputs alone typically did not produce a supra-threshold output (Figure 5A)’ – I think this is shown in the top right corner of the figure, but there is no annotation on the figure and no explanation in the legend.

The values presented in panels 5B-D are difficult to compare between different stimulation patterns. What does success rate correspond to? I assume it is spike probability per EPSG, but this is not stated.

Figure 6 – The spiking plots in Figure 6A are too squashed to be interpretable. Would an additional zoomed-in inset be helpful here?

This figure emphasises the effect of injected synaptic conductances on the membrane potential (line 332) but what about the membrane time constant? Presumably this is equally or more important for adjusting the timing of output generation.

Overall, the interpretation of the results at different frequencies is not really explored. There are two data points at 0 mV for each stimulation frequency. Perhaps one point represents IPSP and the other EPSP? However, I do not understand why they should produce different values.

At the last moment, the study adds in the influence of inhibitory conductances, but the different influences upon neuron output are not really covered in detail. Presumably in vivo, a mixed E/I synaptic conductance is active. How do EPSPs and IPSPs shape neuronal activity in concert?

5. Direct recording of calyx and non-calyx activation

It would be interesting to see the results of direct stimulation of both calyceal and non-calyceal inputs, compared to the dynamic clamp data. Is there a reason why this could not be performed in brain slices from the gerbil?

Minor points:

Line 305 – ‘extend’ -> ‘extent’

Lines 263 and 275 – Figure 2G should be Figure 1G

Reviewer #3

(Remarks to the Author)

Console-Meyer et al.

Non-calyceal inputs gate the timing of calyx of Held evoked MNTB output

Overall:

This manuscript presents a comprehensive and technically rigorous investigation of the synaptic properties of calyceal (calyx of Held) and non-calyceal inputs to principal neurons of the medial nucleus of the trapezoid body (MNTB) in two species. The study addresses an important gap in our understanding of auditory brainstem circuitry by characterizing the previously understudied non-calyceal inputs and examining how these two input types work in concert to modulate neuronal output during sustained activity. This is a meticulously executed brain slice / patch-clamp study that demonstrates clear technical expertise. The experimental design is sound, the recordings are of high quality, and the data are extensive and convincing, providing a thorough characterization of both input types across multiple experimental paradigms. The inclusion of comparative data from two species strengthens the generalizability of the findings. The analytical approaches are appropriate for the data collected, and the statistical methods appear valid and properly applied. The finding that these two input types exhibit opposing forms of short-term plasticity represents an important advance in understanding auditory brainstem computation, and by focusing on the collaborative function of calyceal and non-calyceal inputs, the authors address a gap in the literature where non-calyceal inputs have been largely overlooked for MNTB function. My comments revolve around some minor technical questions but also about the overall interpretations of the data.

Major comments:

1. The authors deserve particular commendation for conducting their experiments in mature animals at such advanced postnatal ages, a methodological decision that decisively eliminates any concerns about developmental confounds in the interpretation of their results. By working with fully mature preparations, the authors ensure that the synaptic properties, short-term plasticity characteristics, and functional interactions they describe genuinely reflect the adult state of this circuit rather than transient developmental features. This approach substantially strengthens the physiological relevance of their findings and allows for more confident extrapolation to the functioning auditory system *in vivo*.
2. The study compares synaptic properties in Mongolian gerbils and Etruscan shrews in an effort to demonstrate that the findings are of general mammalian relevance. While Mongolian gerbils represent an established and appropriate choice given their widespread use in auditory research, including numerous patch-clamp and synaptic studies from this laboratory and others, the selection of Etruscan shrews requires more thorough justification beyond the observation that these species are separated by 94 million years of evolutionary divergence. It would be nice to put the findings in the context of auditory biology and ecological context of the shrew, including its hearing capabilities, lifestyle, and whether it represents a hearing specialist or generalist. Understanding how dependent each species is on precise MNTB-mediated inhibition for behaviors such as sound localization would provide important context for interpreting species differences. The manuscript currently glosses over some notable distinctions between these species, such as the observation that calyceal inputs appear to be two- to three-fold larger in gerbils, which may explain why gerbil inputs exhibit more pronounced depression during train stimulation compared to shrews (Figure 1A versus 1B).
3. As the authors acknowledge, the MNTB represents a more complex synaptic environment than simply calyceal and non-calyceal inputs, which renders some of the overall conclusions somewhat overstated. While the study is clearly executed with great care and the results themselves are entirely convincing and reliable, the broader conclusion that short-term plasticity of these two input types drives temporal reliability warrants more cautious interpretation. The experimental approach necessarily presents a highly reduced view of this brain area, and this reductionism carries the inherent risk that extrapolations from data acquired under these constrained conditions may offer a distorted or at least substantially limited perspective of how the system operates in its full biological context. The MNTB integrates multiple sources of synaptic input, including inhibitory and neuromodulatory inputs. Therefore circuit-level dynamics are difficult to capture in the acute slice preparation, and the interplay between calyceal and non-calyceal inputs likely occurs within this richer computational landscape. At the very least, a more circumspect framing of the conclusions that explicitly acknowledges these experimental limitations would strengthen the manuscript. Beyond that, the inclusion of neuronal modeling might be the simplest approach to capture these additional dynamics.

Minor Comments:

Line 189/190: This statement need clarification. What was the criterion for deciding that amplitudes and decay times could be temporally separated?

Line 216-221: Please rephrase since this does not accurately reflect the results: "... were larger for intermediate frequencies..." since significance is shown for 100 and 300 Hz only.

Line 319-321: Pls clarify the numbers in the bracket. Are units missing? Is the periods decimal signs or a three-digit separators?

Line 337-338: Which groups are compared for the statistical significances presented?

Version 1:

Reviewer comments:

Reviewer #2

(Remarks to the Author)

The authors have responded to the reviews in a satisfactory manner.

One small adjustment: x label of Fig. 5B and 5F would preferentially be 'action potential probability'.

Reviewer #3

(Remarks to the Author)

I have read the revised manuscript and the authors' responses. Overall, I am satisfied with the revision.

My overall assessment remains very positive. Non-calyceal inputs to the MNTB have been known for years, yet nobody has seriously attempted to assign them a functional role. This manuscript does exactly that, and goes further by testing the proposed scenario experimentally. That combination of conceptual novelty and experimental follow-through is what makes this work a genuine advance for the field.

The authors have addressed my major concerns. The comparative species data is now properly contextualized, and a number of overstated conclusions have been sensibly walked back. The new experiment combining direct fiber stimulation with conductance-clamp templates of calyx input is a welcome addition, since it moves things closer to a naturalistic

activation of the circuit, and the results align nicely with the earlier data.

The decision not to pursue modeling is justified. Given that the origin of non-calyceal inputs remains unresolved, a model would rest on too many speculative assumptions to add much.

One caveat worth keeping in mind: these are slice data from a reduced preparation, and the conclusions should be read with that in mind. The authors now acknowledge this openly in the Discussion, clarifying this point for readers of this manuscript.

We hope you will find the referees' comments useful as you decide how to proceed. Should further experimental data or analysis allow you to address these criticisms, we would be happy to look at a substantially revised manuscript. In particular, we agree with Reviewer #2 that "a key experiment appears to be missing – i.e. combining direct activation of both calyceal and non-calyceal inputs in the same neuron, and measuring action potential output." In addition, Reviewer #3 correctly stated that "the MNTB integrates multiple sources of synaptic input, including inhibitory and neuromodulatory inputs", and therefore suggested that a modelling approach could help to "capture these additional dynamics." We are committed to providing a fair and constructive peer-review process. Do not hesitate to contact us if you wish to discuss the revision or if there are specific requests from the reviewers that you believe are technically impossible or unlikely to yield a meaningful outcome.

We thank the reviewers for their constructive and helpful comments. We substantially revised our manuscript and added data from a new experiment. In this experiment, we combined fiber stimulation of the non-calyx inputs with conductance templates of calyx of Held inputs, similar to what reviewer 2 has proposed. The data clearly corroborate and match the previous experimental design of using conductance templates alone. We have justified the choice of animals as both reviewers have suggested. We added to the discussion the limitations of our approach in comparison to *in vivo* data as reviewer 3 requested. We refrain from setting up model predictions, because in our opinion building a biophysical model or in-depth circuit model would clearly exceed this manuscript. Moreover, computational models also contain many assumptions that might not fully match the *in vivo* situation, and in this case the origin of the modulatory, non-calyceal inputs is not fully resolved and a specific stimulus driven response to these inputs is speculative. So, we fairly choose to add considerations about the limitation of our work regarding the transfer to the *in vivo* situation.

Reviewer #2 (Remarks to the Author):

Principal neurons in medial nucleus of the trapezoid body (MNTB) receive excitatory input via the Calyx of Held, a huge excitatory synapse that has served as an important model to study many aspects of synaptic function. The Calyx is built for speed and reliability, underlying its role in sound localisation. However, in addition to calyceal inputs, MNTB neurons also receive additional excitatory and inhibitory inputs across their soma and dendrites. Comparatively little is known about these inputs, and this is where the work of Console-Meyer et al. provides new information. The authors here report the properties of non-calyceal excitatory inputs, including their history-dependence, and propose their functional role may be to improve the reliability and speed up the response timing of MNTB neurons during calyceal activation.

To do this, the authors separately recorded calyx and non-calyx excitatory synaptic input. They then used dynamic clamp to mimic synaptic activation and measure the influence over neuronal output. Overall the study provides new information (particularly demonstrating facilitation of non-calyceal synaptic input). However, a key experiment appears to be missing – i.e. combining direct activation of both calyceal and non-calyceal inputs in the same neuron, and measuring action potential output. The work is statistically sound and repeatable. In general, I found the significance and justification to be lacking for some of the experiments. In its current form, the paper may not be accessible to a general biological readership. I outline these issues below:

1. Comparison between species

It would be helpful to justify the choice of species, and relevance of comparing gerbils and Etruscan shrews.

We added information and justification of the species used in the method and results section. We write in the method section: “In both age groups hearing is fully established and no substantial age-related detrimental effects for hearing are expected. In our preliminary auditory brainstem response data age-related detrimental effects were mildly apparent in Etruscan shrew older 18 months compared to 6-12 months. While Mongolian Gerbils serve as a common model in hearing research, a comparative approach illustrates generality. Etruscan shrews are exquisite hunters that rely next to touch on hearing to detect their prey even at night and therefore require precise spatial information from the auditory system. In both gerbils and Etruscan shrew, the medial nucleus of the trapezoid body is well developed and contains large somatic synapses. MNTB mediated inhibition is relevant at least for sound source localization targeting MSO and LSO. In Etruscan shrew no clear structure MSO is detectable (51) consistent with their exclusive high frequency listening above 6 kHz. Thus, in comparison to gerbils, which hear between 0.5 and 50 kHz, Etruscan shrews are specialists for high frequency listening.”

We have also revised a sentence in the results: “Comparative recordings in gerbils and Etruscan shrews were carried out to highlight the general attributes of STP in both synapse types in this archaic neuronal circuit. In both species, non-calyceal inputs rather exhibit STF in contrast to dominating STD of the calyx of Held (Figure 1A and B), indicating an overall evolutionary stability of STP.”

Brain slices of different maturity were used for the two species (adolescent gerbils and adult shrews). Is this a confounding or a deliberate part of the experimental design? Potentially this difference could account for differences in EPSC amplitude and overall synaptic facilitation between the two groups. While there are certainly notable differences in amplitude and dynamics of calyx and non-calyx EPSCs, it is not clear how seriously should we compare differences between gerbil and shrew data given the age differences. Since the shrew data is dropped after Figure 2, it is not clear to me how important is in to the overall paper.

The age ranges might insinuate a substantial difference in the maturation of the animals. However, the auditory system can be considered readily developed in gerbils from postnatal day 22 on. Later in the gerbil's live span, approximately 1.5 to 2 years of age, the auditory system starts to deteriorate. Since the early auditory brainstem system is rather independent of hormonal regulation substantial changes in the adolescent period are not expected in gerbils. In Etruscan shrew no developmental study has been conducted, the postnatal onset of hearing is unknown. We know from our preliminary auditory brainstem response data that hearing does not or only mildly deteriorate in aged animals >1.5 years. To work in both animals with developed, functional and comparable hearing we have chosen the presented age ranges. We included the following sentence for better justification: “In both age groups hearing is fully established and no substantial age-related detrimental effects for hearing are expected. In our preliminary auditory brainstem response data age-related detrimental effects were mildly apparent in Etruscan shrew older 18 months compared to 6-12 months.”

Comparative research is a crucial approach to validate evolutionary stability or variations of biological systems. We consider it important to show that unexplored inputs are ubiquitous amongst mammals and have similar functions. We have added a sentence that supports the usage and importance of our approach: “Comparative recordings in gerbils and Etruscan shrews were carried out to highlight the general attributes of STP in both synapse types in this archaic neuronal circuit. In both species, non-calyceal inputs rather exhibit STF in contrast to dominating STD of the calyx of

Held (Figure 1A and B), indicating an overall evolutionary stability of STP.”. In addition, these comparative data are interesting in the light of recent findings that report similar differences in the calyx of Held mediated EPSCs between gerbils and bats. Thus, suggesting that in rodents this synaptic junction has specific adaptations. Therefore, we state in the discussion: “Alternatively, like bats, Etruscan shrews might be more adapted to ongoing sounds than gerbils (28) or gerbils and other rodents show a specific adaptation at this synaptic junction.” Thus, including the Etruscan shrew data about the synaptic calyceal and non-calyceal inputs is relevant for our understanding of this archaic nucleus.

Line 109 – ‘In both species, non-calyceal inputs exhibit strong STF in contrast to strong STD of the calyx...’

Has been changed to: “In both species, non-calyceal inputs rather exhibit STF in contrast to dominating STD of the calyx of Held (Figure 1A and B), indicating an overall evolutionary stability of STP.”

Currently this is not easy to determine from the data, especially in the gerbil (Fig 1A,1E). Perhaps this could be made more convincing by zooming in on the first few pulses of the stimulus train? In the gerbil, high stimulation frequencies seem to be subject to neither facilitation or depression. For clarification we have added a dotted line at value 1 to figure 1E and state in the figure legend: “Values below the dotted line indicate depression, values above facilitation.” This line is consistent with Figure 1G. Together with the revised sentence (above), we consider this statement softened and better illustrated. A zoom in for each of the traces in figure 1a was not really instructive and made the figure even more dense. Therefore, we decided that the addition of the dotted line was the best choice.

2. Voltage clamp recordings of calyceal vs non-calyceal EPSCs

Is it problematic that a different internal solution was used to measure STP for calyceal vs non-calyceal input? Presumably recordings with the potassium-based intracellular solution will have higher noise. Also if non-calyceal input is arriving on the dendrite, could there be a space clamp issue, exacerbated by use of a potassium-based solution? Both possibilities could impair measure of evoked non-calyceal EPSCs.

We consider the differential use of internal solutions as not critical, rather as appropriate. First, comparing the results from Figure 2 and Figure 3 shows the expected results that larger and broader responses that do not depress can be recorded under elevated calcium concentrations. Thus, the potassium and cesium clamp seem to match the expectations. Second, the input resistance of gerbil MNTB neurons of 60-90 MOhm and in Etruscan shrew of about 150-200 MOhm is high enough to achieve good clamp conditions without additional blockage of ion channels. Especially, as the synaptic responses are small a local voltage escape is not expected. Shure space clamp effects from far distal dendrites will always occur. However, we are not certain at which position the non-calyceal inputs arrive to the MNTB neurons. This is why we have not specified them as exclusively “dendritic inputs” but as “non-calyceal inputs”. However, by observing a small correlation between EPSC decay and size it could be assumed that dendritic filtering exists. It is crucial to record calyx responses with cesium-based solutions. Because of the rapid rise and size of these EPSCs even a local voltage escape can occur, triggering sodium currents. Thus, the adjustments of internal recording solutions are appropriate. Overall, we consider the use of both solutions as valid.

3. Vesicle refilling dynamics

The significance of studying the vesicle refilling dynamics of non-calyceal synapses could be made more accessible. Was there a specific hypothesis here and what is the physiological relevance? Perhaps data from calyx synapses could be overlaid in some way on Figure 3D so the comparison can be made obvious, and support the claim that calyceal and non-calyceal inputs exhibit the same dynamics.

We have rephrased the motivation of this section to clarify why we have performed and included these experiments. The paragraph reads now: "Because the calyx of Held and the non-calyceal inputs display dissimilar STP forms, we reasoned that the vesicular refilling dynamics and calcium dependence of the STP phenotype might differ between both synaptic input types. Since the vesicle dynamics of gerbil calyx of Held synapses have been elucidated (30, 39) and the calcium-dependent shift from STD to facilitation of the calyx synapse is well documented, we focused on the non-calyceal inputs in gerbils to allow for a direct comparison. To determine calcium dependency of STP and the refilling time course, vesicle pool depletion was enhanced by an elevation in extracellular calcium concentration, which is considered to increase the release probability and thus benefits pool depletion." We decided not to overlay the data from Figure 3D with existing data from the literature. However, for better direct comparison with the literature, we have changed the X-axis scaling to closer match those in the literature.

4. Functional significance demonstrated with dynamic clamp

Figure 5 – I have a hard time understanding this figure. Mainly, it is difficult to assess differences in responses to conditioned and non-conditioned inputs – for example subpanels a and c look almost identical in main Panel A. At present, it is difficult to get a sense of how non-calyceal conductances contribute to the neuron output.

We have reorganized this figure to improve visualization and simplicity. Instead of overlaying too many conditions we have split the data. The first part of the figure illustrates the input-output functions based on unconditioned non-calyx and calyx templates (Figure 5A-D), while the second part shows the data from the templates that correlate to the conditioned non-calyx and calyx inputs (Figure 5E-H). The labelling of the figure was revised so that the specification is correct now. The raw data is more directly linked to the data representation. We have revised the section in the manuscript accordingly. We hope that this improves the readability of this figure and facilitates understanding of the experiment.

Line 293 – 'Non-calyceal inputs alone typically did not produce a supra-threshold output (Figure 5A)' – I think this is shown in the top right corner of the figure, but there is no annotation on the figure and no explanation in the legend.

This sentence has been revised and reads now: "Conductance of non-calyceal inputs alone typically did not produce a supra-threshold output and were not considered further for the analysis of suprathreshold output generation." Since this data was not used for analysis in figure 5 we have removed the raw data from figure 5.

The values presented in panels 5B-D are difficult to compare between different stimulation patterns. What does success rate correspond to? I assume it is spike probability per EPSP, but this is not stated. The reviewer is correct. We have plotted the action potential probability. We have renamed the Y-axis accordingly. For better comparison we have split the data into two graphs (5A-D vs. 5E-H). The figure legends have been revised according to the new version of figure 5.

Figure 6 – The spiking plots in Figure 6A are too squashed to be interpretable. Would an additional zoomed-in inset be helpful here?

We have not only added new data to this figure but have also rearranged the data. For better visibility

we have stretched the spiking plots in the former figure 6A (now Figure 6E and F). This rearrangement allows seeing single action potentials and therefore an additional zoom in appears unnecessary.

This figure emphasises the effect of injected synaptic conductances on the membrane potential (line 332) but what about the membrane time constant? Presumably this is equally or more important for adjusting the timing of output generation.

The reviewer raises an interesting point. We have plotted the latency values against the membrane time constant and the membrane resistance. These graphs are given below, where the different symbols represent 100 (open) and 400 Hz (closed) calyx template stimulation frequency and the different colors the intensity of the excitatory, modulatory, steady-state conductance. From these graphs no correlation between the latency and membrane time constant or input resistance is apparent. We would argue that the membrane time constant and input resistance are more relevant for sub-threshold integration and less for spike timing. This might be especially the case when a steady-state conductance and not a rapid fluctuating stimulation is applied. Under steady-state conditions close to action potential threshold, the membrane time constant might be less relevant and the input resistance at resting levels overwritten by activated conductances and the additional leak from the steady state conductance template. We find this interesting but consider it not central to this manuscript and have decided not to pursue this analysis and add data to the manuscript.

Overall, the interpretation of the results at different frequencies is not really explored. There are two data points at 0 mV for each stimulation frequency. Perhaps one point represents IPSP and the other EPSP? However, I do not understand why they should produce different values.

We thank the reviewer for identifying this inaccuracy. We first revised the data and second added a sentence for clarification. The sentence states: "The conductance of 0 was applied for both excitatory and inhibitory modulation and hence responses overlapped." When reviewing the data, we realized that the number of included cells for the excitatory and inhibitory modulatory inputs were not the same within a given frequency of calyx stimulation. We therefore eliminated cells that were only stimulated either with inhibitory or excitatory modulatory conductances. After removing these responses, the "overlapping" 0 conductance condition from excitatory and inhibitory modulations fell on top of each other, as they should. The statistics were reviewed and the new numbers are given. These revised statistics showed the same significances as before.

At the last moment, the study adds in the influence of inhibitory conductances, but the different influences upon neuron output are not really covered in detail. Presumably in vivo, a mixed E/I synaptic conductance is active. How do EPSPs and IPSPs shape neuronal activity in concert?

We have included inhibitory action on the temporal precision of output generation to show the full width of time shifts. We consider this important as the inhibitory input has been documented and shows similar STP (Mayer, Klug, *J Neurophysiol*; 2014) compared to the non-calyceal excitatory inputs. We agree that under *in vivo* conditions, non-calyceal excitatory and inhibitory inputs are possibly present at the same time. How they interact depends on their temporal appearance and intensity. Because it is unknown how sound or other stats/neuromodulations will drive these non-calyceal inputs, we cannot even speculate about their generated E/I conductance. Therefore, we found it appropriate to restrict our experiments to the simple, reduced situation of a single type of input to illustrate the possible output-dynamics of this system, since these are fundamental for the circuitry.

5. Direct recording of calyx and non-calyx activation

It would be interesting to see the results of direct stimulation of both calyceal and non-calyceal inputs, compared to the dynamic clamp data. Is there a reason why this could not be performed in brain slices from the gerbil?

We have added a new experiment where afferent non-calyx inputs were electrically stimulated and combined with calyx-like conductance templates. The results match the conductance clamp recording where both inputs were simulated with conductance templates. We have chosen this experimental design for the following reasons. Calyx inputs arrive at the soma and can be simulated by conductance without spatial distortion from the somatic patch pipette. Simulating calyx inputs allows for a stable latency control without the synaptic imposed fluctuations, reducing the variability in the recording and allowing to test more conditions instead of running more repetitions. In addition, the depression of the calyx of Held synapse can be ruled out to contribute to the effect. We present these data now in figure 6 in combination with the previous data.

Minor points:

Line 305 – ‘extend’ -> ‘extent’

Has been changed accordingly

Lines 263 and 275 – Figure 2G should be Figure 1G

Has been changed accordingly

Reviewer #3 (Remarks to the Author):

Console-Meyer et al.

Non-calyceal inputs gate the timing of calyx of Held evoked MNTB output

Overall:

This manuscript presents a comprehensive and technically rigorous investigation of the synaptic properties of calyceal (calyx of Held) and non-calyceal inputs to principal neurons of the medial nucleus of the trapezoid body (MNTB) in two species. The study addresses an important gap in our understanding of auditory brainstem circuitry by characterizing the previously understudied non-calyceal inputs and examining how these two input types work in concert to modulate neuronal output during sustained activity. This is a meticulously executed brain slice / patch-clamp study that demonstrates clear technical expertise. The experimental design is sound, the recordings are of high quality, and the data are extensive and convincing, providing a thorough characterization of both input types across multiple experimental paradigms. The inclusion of comparative data from two species strengthens the generalizability of the findings. The analytical approaches are appropriate for the data

collected, and the statistical methods appear valid and properly applied. The finding that these two input types exhibit opposing forms of short-term plasticity represents an important advance in understanding auditory brainstem computation, and by focusing on the collaborative function of calyceal and non-calyceal inputs, the authors address a gap in the literature where non-calyceal inputs have been largely overlooked for MNTB function. My comments revolve around some minor technical questions but also about the overall interpretations of the data.

Major comments:

1. The authors deserve particular commendation for conducting their experiments in mature animals at such advanced postnatal ages, a methodological decision that decisively eliminates any concerns about developmental confounds in the interpretation of their results. By working with fully mature preparations, the authors ensure that the synaptic properties, short-term plasticity characteristics, and functional interactions they describe genuinely reflect the adult state of this circuit rather than transient developmental features. This approach substantially strengthens the physiological relevance of their findings and allows for more confident extrapolation to the functioning auditory system *in vivo*.

2. The study compares synaptic properties in Mongolian gerbils and Etruscan shrews in an effort to demonstrate that the findings are of general mammalian relevance. While Mongolian gerbils represent an established and appropriate choice given their widespread use in auditory research, including numerous patch-clamp and synaptic studies from this laboratory and others, the selection of Etruscan shrews requires more thorough justification beyond the observation that these species are separated by 94 million years of evolutionary divergence. It would be nice to put the findings in the context of auditory biology and ecological context of the shrew, including its hearing capabilities, lifestyle, and whether it represents a hearing specialist or generalist. Understanding how dependent each species is on precise MNTB-mediated inhibition for behaviors such as sound localization would provide important context for interpreting species differences. The manuscript currently glosses over some notable distinctions between these species, such as the observation that calyceal inputs appear to be two- to three-fold larger in gerbils, which may explain why gerbil inputs exhibit more pronounced depression during train stimulation compared to shrews (Figure 1A versus 1B).

We have extended the method section about the animals to give more background information and justification for the choice of species “In both age groups hearing is fully established and no substantial age-related detrimental effects for hearing are expected. In our preliminary auditory brainstem response data age-related detrimental effects were mildly apparent in Etruscan shrew older 18 months compared to 6-12 months. While Mongolian Gerbils serve as a common model in hearing research, a comparative approach illustrates generality. Etruscan shrews are exquisite hunters that rely next to touch on hearing to detect their prey even at night and therefore require precise spatial information from the auditory system. In both gerbils and Etruscan shrew, the medial nucleus of the trapezoid body is well developed and contains large somatic synapses. MNTB mediated inhibition is relevant at least for sound source localization targeting MSO and LSO. In Etruscan shrew no clear structure MSO is detectable (51) consistent with their exclusive high frequency listening above 6 kHz. Thus, in comparison to gerbils, which hear between 0.5 and 50 kHz, Etruscan shrews are specialists for high frequency listening.” We have also revised a sentence in the results: “Comparative recordings in gerbils and Etruscan shrews were carried out to highlight the general attributes of STP in both synapse types in this archaic neuronal circuit. In both species, non-calyceal inputs rather exhibit STF in contrast to dominating STD of the calyx of Held (Figure 1A and B), indicating an overall evolutionary stability of STP.”

To give more weight on the different calyx EPSC sizes between gerbils and Etruscan shrews we have added information in the discussion: “One explanation for the EPSC size difference might be a

different initial release probability, in agreement with the larger STD in gerbils compared to Etruscan shrew. The molecular and cellular mechanisms for a possible different initial release probability are so far unclear. Functionally, the EPSC size reduction in Etruscan shrew might be adapted to match the so far unexplored postsynaptic properties of these overall smaller neurons. Alternatively, like bats, Etruscan shrews might be more adapted to ongoing sounds than gerbils (28) or gerbils and other rodents show a specific adaptation at this synaptic junction”

3. As the authors acknowledge, the MNTB represents a more complex synaptic environment than simply calyceal and non-calyceal inputs, which renders some of the overall conclusions somewhat overstated. While the study is clearly executed with great care and the results themselves are entirely convincing and reliable, the broader conclusion that short-term plasticity of these two input types drives temporal reliability warrants more cautious interpretation. The experimental approach necessarily presents a highly reduced view of this brain area, and this reductionism carries the inherent risk that extrapolations from data acquired under these constrained conditions may offer a distorted or at least substantially limited perspective of how the system operates in its full biological context. The MNTB integrates multiple sources of synaptic input, including inhibitory and neuromodulatory inputs. Therefore circuit-level dynamics are difficult to capture in the acute slice preparation, and the interplay between calyceal and non-calyceal inputs likely occurs within this richer computational landscape. At the very least, a more circumspect framing of the conclusions that explicitly acknowledges these experimental limitations would strengthen the manuscript. Beyond that, the inclusion of neuronal modeling might be the simplest approach to capture these additional dynamics.

We have scaled down some of the conclusions to not overstate the interpretation of our data. Examples are:

Line 109: originally: “and thereby affecting sound processing” to “and thereby might affect sound processing”

Line 120: originally: “non-calyceal inputs exhibit strong STF in contrast to the STD of the calyx of Held” to “non-calyceal inputs rather exhibit STF in contrast to dominating STD of the calyx of Held”

Line 472: originally: “we can determine the shift” to “we can estimate a potential the shift”

Line 477: originally: “shift is supposed to functionally” to “shift should be capable to functionally”

Line 451: originally: “The strong asynchronous” to “The substantial asynchronous”

In addition, we have added a less artificial experiment, where we stimulated the afferent, non-calyceal inputs and paired them with the simulated calyx of Held EPSC. These data are presented in figure 6. The new results fit very well with the data we have obtained from the previous dynamic clamp recordings. Certainly, all slice work is reductionistic and therefore limited and partially artificial. In general, in slice experiments it is not possible to generate the native environment with its full neuromodulation and ongoing background activity driven by sensory systems. However, the essence of synaptic interactions can be studied at its best. We argue that modeling is artificial as well, as it also represents a reductionistic approach, where estimated assumptions are required. Sure, a model can provide additional support to the conclusions and statements for the circuit level. However, a detailed biophysical model for the synaptic interactions in MNTB neurons or a full-scale circuit model for sensory inputs and outputs from the MNTB even only to the binaural coincidence detectors is beyond the reach of this manuscript. Especially since the origin of non-calyceal inputs is not fully resolved and therefore

their stimulus driven activity pattern would be largely speculative. We agree that different modelling approaches that will use our findings will provide valuable support and insights in the future.

We now acknowledge the limitations of our results in respect to the interpretation in the *in vivo* situation by stating in the discussion: “However, to translate our *in vitro* findings, acquired in an experimental environment of limited modulation, background and no sensory stimulation, to the *in vivo* situation remains limited and speculative”

Minor Comments:

Line 189/190: This statement need clarification. What was the criterion for deciding that amplitudes and decay times could be temporally separated?

We have rephrased this sentence and added more information: “When delayed EPSCs did not overlap but appeared temporally separated, their amplitudes and decay times were individually extracted (Figure 2A and B).” and “To analyze whether the EPSC amplitudes and decay times within the delayed release events differ between gerbils and Etruscan shrew, individual EPSC events of delayed release were isolated and their amplitude was determined by finding the minimum peak. From that minimum, we further fitted a biexponential curve to calculate individual EPSC decay times. The extracted events from one recording were pooled, their average and standard deviation were determined”

Line 216-221: Please rephrase since this does not accurately reflect the results: “... were larger for intermediate frequencies...” since significance is shown for 100 and 300 Hz only. We have rephrased this sentence to: “Moreover, the number of detected delayed EPSCs appeared larger in gerbil compared to shrew at least for some frequencies...”

Line 319-321: Pls clarify the numbers in the bracket. Are units missing? Is the periods decimal signs or a three-digit separators?

We thank the reviewer for spotting the lack of units. The values represent nS. For clarity the digits are now cut down to one.

Line 337-338: Which groups are compared for the statistical significances presented?

We did not compare statistical differences between frequencies, because we were interested whether the observed shift due to modulatory effects of either additional excitatory conductance or inhibitory conductance is significant. Therefore, we compared each averaged data point shown for inhibition and excitation within one stimulation frequency via Friedman-Test, since these data are not normally distributed.